# Exploration-free Algorithms for Multi-group Mean Estimation

**Ziyi Wei** [1]  **Huaiyang Zhong** [1]  **Xiaocheng Li** [2]

## Abstract

We address the problem of multi-group mean estimation, which seeks to allocate a finite sampling budget across multiple groups to obtain uniformly accurate estimates of their means. Unlike classical multi-armed bandits, whose objective is to minimize regret by identifying and exploiting the best arm, the optimal allocation in this setting requires sampling every group on the order of $\Theta(T)$ times. This fundamental distinction makes exploration-free algorithms both natural and effective. Our work makes three contributions. First, we strengthen the existing results on subgaussian variance concentration using the Hanson-Wright inequality and identify a class of strictly subgaussian distributions that yield sharper guarantees. Second, we design exploration-free non-adaptive and adaptive algorithms, and we establish tighter regret bounds than the existing results. Third, we extend the framework to contextual bandit settings, an underexplored direction, and propose algorithms that leverage side information with provable guarantees. Overall, these results position exploration-free allocation as a principled and efficient approach to multi-group mean estimation, with potential applications in experimental design, personalization, and other domains requiring accurate multi-group inference.

## 1. Introduction

We study the problem of multi-group mean estimation, where the task is to allocate a limited sampling budget across multiple groups in order to estimate their means uniformly well. This problem arises naturally in polling, survey design, marketing, experimental design, adaptive A/B testing, and other settings where representative estimates across diverse groups are required. A particularly important example is average treatment effect (ATE) estimation in randomized experiments. In this setting, the standard difference-in-means estimator reduces the statistical objective to estimating the mean outcomes of treatment and control groups, while the design problem becomes how to allocate samples across groups so as to minimize estimation error. This perspective is closely related to classical Neyman allocation and its adaptive variants for experimental design and treatment-effect estimation (Zhao, 2024; Li et al., 2024), with representative applications in social science experiments and multi-treatment experimental design (Blackwell et al., 2023; Xu et al., 2025).

A key feature distinguishing this setting from classical reward-maximization bandits is that the optimal allocation requires sampling every arm on the order of $\Theta(T)$ times, rather than focusing as much as possible on the best option. This structural property suggests that explicit exploration phases are unnecessary and opens the door to exploration-free algorithms.

Contextual information makes the problem even more relevant in real-world applications such as healthcare (Bastani & Bayati, 2020; Du et al., 2024), recommendation systems (Agarwal et al., 2009; Li et al., 2010), and dynamic pricing (Qiang & Bayati, 2016; Ban & Keskin, 2021), where side information fundamentally shapes the reward distributions and motivates the estimation of context-dependent group parameters. Accurate estimation in this richer setting is crucial for interpretable personalization, robust policy design, and fairness considerations.

**Literature Review.** In the traditional bandit setting, several groups of papers have studied this problem and proposed several extensions on the objective metric (Antos et al., 2008; 2010; Carpentier et al., 2011; Shekhar et al., 2020; Aznag et al., 2023). For linear models, in particular linear and contextual bandits, the central task reduces to estimating the unknown coefficient vector $\beta$. Riquelme et al. (2017) analyzed this problem in the multi-linear regression setting under the assumption that both the noise and the context vectors are Gaussian. Fontaine et al. (2021) focused on the univariate case of $\beta^*$ and evaluated performance using the metric $\mathbb{E}[\|\hat{\beta} - \beta^*\|^2]$, their analysis accommodates heterogeneous subgaussian noise while assuming that the context vectors have unit norm. Beyond these closely re-

---

[1]Grado Department of Industrial and Systems Engineering, Virginia Tech [2]Imperial College Business School, Imperial College London. Correspondence to: Ziyi Wei <ziyiw@vt.edu>.

*Proceedings of the 43rd International Conference on Machine Learning*, Seoul, South Korea. PMLR 306, 2026. Copyright 2026 by the author(s).

lated works, the literature on group mean estimation also connects to research in areas such as best-arm identification, conservative bandits, and experimental design. We defer more discussions and other related literature to Appendix A.

**Contributions.** Our contributions are threefold: First, we strengthen the existing results on subgaussian variance concentration using the Hanson-Wright inequality and identify a class of strictly subgaussian distributions that yield sharper guarantees. Second, we design non-adaptive and adaptive algorithms that are exploration-free, and we establish tighter regret bounds than the existing results. Third, we extend the framework to contextual bandit settings, an underexplored direction, and propose algorithms that leverage side information with provable guarantees. Theoretically, our results reveal certain structural properties of the problem, such as exploration-free design, the failure mode of UCB-type algorithms for the contextual setting, and also connections with the best-arm identification problem.

## 2. Problem Setup

In this section, we present the problem setup of *multi-group mean estimation*. Consider $K$ alternatives (also known as arms in the multi-armed bandits literature), each of which is associated with a random outcome/reward. Specifically, the outcome of $k$-th alternative follows a distribution $\mathcal{P}_k$ with unknown mean $\mu_k$ and variance $\sigma_k^2$. We consider an online learning setup where there is a finite horizon $T$. At each time $t = 1, ..., T$, the decision maker chooses one alternative $k_t \in \{1, ..., K\}$, and then (s)he observes and only observes the outcome of a realization $X_{k_t} \sim \mathcal{P}_k$. Notation-wise, if we take the standpoint of each alternative $k$, we use $X_{k,t}$ to denote the $t$-th observations we collect from the $k$-th alternative. Accordingly, we use $n_k$ to denote the total number of times the $k$-th alternative is chosen throughout the horizon $T$. In this way, we can estimate the mean of alternative $k$ by the average of the observations

$$\hat{\mu}_k(T) := \frac{1}{n_k} \sum_{t=1}^{n_k} X_{k,t}.$$

The task of *multi-group mean estimation* aims to accurately estimate the mean for all alternatives simultaneously. By the end of the horizon, the estimation error for alternative $k$ is

$$\mathbb{E}_{\mathcal{P}_k}[(\hat{\mu}_k(T) - \mu_k)^2] = \frac{\sigma_k^2}{n_k}. \tag{1}$$

To aggregate the errors across all the alternatives, Aznag et al. (2023) propose the following objective for multi-group

mean estimation, for $p > 0$,

$$R_p(\boldsymbol{n}) := \left\| \left\{ \frac{\sigma_k^2}{n_k} \right\}_{k=1}^K \right\|_p = \begin{cases} (\sum_{k=1}^K \frac{\sigma_k^{2p}}{n_k^p})^{\frac{1}{p}}, & \text{if } p < \infty, \\ \max_{k=1,...,K} \frac{\sigma_k^2}{n_k}, & \text{if } p = \infty. \end{cases}$$

We note the objective only involves $\boldsymbol{n} = (n_1, ..., n_K)$, the number of observations for each alternative, as the decision variables. Intuitively, the objective requires budgeting the observations in a way that we obtain a uniformly good mean estimation for all the alternatives. Importantly, the variance $\sigma_k^2$ is unknown and has to be estimated from the observations as well.

**Proposition 2.1.** *Suppose one knows $\sigma_k^2$, then the optimal allocation scheme $\boldsymbol{n}^* = (n_1^*, ..., n_K^*)$ is given by*

$$n_k^* = \frac{\sigma_k^q}{\sum_{j=1}^K \sigma_j^q} \cdot T,$$

*where $q := \frac{2p}{p+1}$ if $p$ is finite and $q = 2$ if $p = \infty$ and the optimal objective value*

$$R_p(\boldsymbol{n}^*) = \frac{1}{T} \left( \sum_{k=1}^K \sigma_k^q \right)^{\frac{2}{q}}.$$

Proposition 2.1 is obtained from solving the optimization problem with the knowledge of $\sigma_k^2$. Without the knowledge, an algorithm can never beat this objective value, and will be benchmarked against the value to measure the algorithm's performance. We also note that the optimal solution depends both on $\sigma_k^2$'s and the norm $p$, and the optimal value scales on the order of $1/T$.

Before we talk about our algorithms, we first introduce some assumptions and basic inequalities to help our analyses. First, we introduce the concept of subgaussian and strictly subgaussian.

**Definition 2.2.** A random variable $X$ with distribution $\mathcal{P}_X$ is $\sigma$-subgaussian such that for all $t \in \mathbb{R}$:

$$\mathbb{E}_{X \sim \mathcal{P}_X}[\exp(tX)] \leq \exp(t^2 \sigma^2 / 2). \tag{2}$$

Moreover, if $X$'s variance is $\sigma_X^2$, we say $X$ is strictly subgaussian if it satisfies (2) with $\sigma^2 = \sigma_X^2$.

**Assumption 2.3.** Throughout our paper, we assume

(a) $\mathcal{P}_k$ follows $\sigma$-subgaussian for all $k$.

(b) $\sigma_{\min}^2 = \min_{k=1,...,K} \sigma_k^2$ is positive.

We also assume $\sigma$ is known.

These are two mild assumptions: Part (a) is commonly assumed in multi-armed bandits literature, and Part (b) simply says that all the alternatives are random (if there is a deterministic one, we don't really need to estimate its mean).

## 2.1. Subgaussian variance concentration

While the optimal allocation scheme is determined by $\sigma_k^2$ as in Proposition 2.1, any algorithm that solves the problem should naturally involve some variance estimation, i.e., estimating $\sigma_k^2$ from observations. So we first state several concentration inequalities related to variance estimation. For this subsection, we state the results for a general random variable $X$.

For $n$ i.i.d. observations $X_1, ..., X_n$, one can construct a variance estimator

$$\hat{\sigma}_n^2 := \frac{1}{n-1} \sum_{i=1}^{n} (X_i - \bar{X})^2, \qquad (3)$$

where $\bar{X}$ is the sample mean. Let $\sigma_X^2$ be the true variance of $X$.

**Lemma 2.4.** *Suppose $X$ is $\sigma$-subgaussian, then we have*

$$\mathbb{P}\left(|\hat{\sigma}_n^2 - \sigma_X^2| \geq 4\sigma^2 f(n) \sqrt{\frac{2\log(1/\delta)}{n-1}} + \frac{6\sigma^2 \log(1/\delta)}{n}\right) \leq 2\delta,$$

*where $f(n) = (1 + \sqrt{n-1})/\sqrt{n}$. Specifically, if $X$ is strictly subgaussian, then $f(n) = (1 + \sqrt{(n-1)/8})/\sqrt{n}$, and $\sigma$ can be replaced with $\sigma_X$.*

For the results, we refer to Appendix B for the proof sketch and a detailed discussion. It utilizes the structure of the variance estimator and gets rid of the term $\sqrt{\log(1/\delta)}$ used in the previous work (Aznag et al., 2023). The lemma tells that the error of the variance estimator shrinks at a rate of $1/\sqrt{n}$. A subtle point is that when $X$ is strictly subgaussian, the constants are improved and the subgaussian parameter $\sigma$ is improved to the true variance. For the case of the Gaussian distribution, the bound can be further tightened based on Lemma 1 in Laurent & Massart (2000):

$$\mathbb{P}\left(\hat{\sigma}_n^2 - \sigma_X^2 \geq 2\sigma_X^2 \sqrt{\frac{\log(1/\delta)}{n-1}} + \frac{2\sigma_X^2 \log(1/\delta)}{n-1}\right) \leq \delta,$$

$$\mathbb{P}\left(\sigma_X^2 - \hat{\sigma}_n^2 \geq 2\sigma_X^2 \sqrt{\frac{\log(1/\delta)}{n-1}}\right) \leq \delta. \qquad (4)$$

For these bounds, the constants of the leading-order term are close to those of the strictly subgaussian case, which suggests that the strictly subgaussian structure offers a level of tail control comparable to the most ideal Gaussian case.

## 3. Non-adaptive-Style Algorithm

We first present a non-adaptive algorithm for the problem. The algorithm doesn't require knowing $\sigma_k^2$ exactly, but requires a knowledge of a lower bound $\underline{\sigma}^2$, i.e. for all $k = 1, ..., K$,

$$\sigma_k^2 \geq \underline{\sigma}^2 > 0.$$

The knowledge of $\underline{\sigma}^2$ can usually be obtained from historical data or domain knowledge. We will fully remove this requirement in the next section. Here we use the setup to generate more intuitions for the algorithm design. It also provides insights into the special structure of multi-group mean estimation and how it differs from multi-armed bandits and best arm identification (Audibert & Bubeck, 2010).

---

**Algorithm 1** Non-adaptive allocation

---

**Require:** $T$, initial length $\tau$, constant $q$
1: **Phase 1: Uniformly select and estimate**
2: **for** each alternative $k = 1, \ldots, K$ **do**
3:     Choose $k$ for $\tau$ rounds (time periods)
4: **end for**
5: Estimate $\hat{\sigma}_{k,\tau}^2$ with (3)
6: **Phase 2: Allocate the remaining periods**
7: Compute allocation weight $\lambda_{k,\tau} = \hat{\sigma}_{k,\tau}^q / \sum_{j=1}^{K} \hat{\sigma}_{j,\tau}^q$
8: **for** each alternative $k = 1, \ldots, K$ **do**
9:     Choose $k$ for $\lambda_{k,\tau} T - \tau$ rounds
10:    Calculate $\hat{\mu}_k(T) = \frac{1}{\lambda_{k,\tau} T} \sum_{t=1}^{\lambda_{k,\tau} T} X_{k,t}$
11: **end for**
12: **Output:** Final estimates $\{\hat{\mu}_k(T)\}_{k=1}^{K}$

---

Algorithm 1 requires two inputs (in addition to the horizon length $T$): the initial length $\tau$ and the constant $q$. First, $q$ is determined by the norm $p$ in the performance measure, and the definition is given in Proposition 2.1. Second, the exploration length $\tau$ is given by the following

$$\tau := \frac{\underline{\sigma}^q}{\underline{\sigma}^q + (K-1) \cdot \sigma^q} \cdot T \qquad (5)$$

where $\underline{\sigma}^2$ is the variance lower bound and $\sigma$ is the subgaussian parameter. It is easy to verify (from Proposition 2.1) that such a choice ensures that

$$\tau \leq \min_{k=1,\ldots,K} n_k^*.$$

For notation simplicity, we just assume all the values are integers and omit the floor symbol. The algorithm is a direct implication of Proposition 2.1. Recall that Proposition 2.1 says the optimal allocation scheme $\boldsymbol{n}^*$ depends on the true $\sigma_k^2$. The algorithm basically estimates the variances with observations in the exploration and then allocates the remaining time periods according to the optimal solution structure in Proposition 2.1. We call the algorithm as *non-adaptive* allocation in that the variances are estimated just based on the initial $\tau$ observations, and then the allocation scheme is determined accordingly and will not be adaptively adjusted later. This non-adaptive nature of the algorithm resembles the non-adaptive design (Glynn & Juneja, 2004) for the best-arm identification problem with known variance.

The algorithm has a simple and intuitive structure. However, we'd like to make a few important remarks. The initial

phase goes in a round-robin manner. We deliberately avoid calling it an *exploration* phase. The reason is that, if we think about the *exploration* in multi-armed bandits literature, it generally refers to certain actions taken to collect data/information in sacrifice of short-term reward. Specifically, any play of suboptimal arms in multi-armed bandits will incur regret, but such plays are inevitable if we want to learn the system. Yet, for the context of multi-group mean estimation, Proposition 2.1 says that the optimal allocation requires going with each alternative $\Omega(T)$ times. Thus, the initial phase of Algorithm 1 is not only to construct variance estimates, but these rounds of selections are indeed necessarily required by the optimal solution. That's why we call our algorithm exploration-free. The $\Omega(T)$ times of selections prescribed by the optimal solution give a sufficiently good estimation of the system, and no additional exploration is needed. In this light, our result tells that the UCB design in Aznag et al. (2023) is redundant. In addition, we want to compare multi-group mean estimation with the problem of best arm identification. For both problems, they have an objective function different from the regret in multi-armed bandits. A special point is that the objective function $R_p(\boldsymbol{n})$ of multi-group mean estimation is closed-form in terms of the allocation scheme $\boldsymbol{n}$, whereas the probability of correct selection has a complicated relation with the allocation scheme (except for simple cases like two-armed bandits). The closed-formedness is the key to admitting simple algorithms like Algorithm 1.

### 3.1. Analysis of Algorithm 1

To facilitate our presentation, we define

$$\Sigma_q := \sum_{k=1}^{K} \sigma_k^q,$$

$$\lambda := \frac{\underline{\sigma}^q}{\underline{\sigma}^q + (K-1) \cdot \sigma^q} = \frac{\tau}{T}$$

where $q$ is determined by $p$ as in Proposition 2.1.

We denote the allocation scheme of Algorithm 1 as $\boldsymbol{n}_{\pi_1}$ that represents the number of times each alternative is selected by the end of the horizon under Algorithm 1. Let $\boldsymbol{\sigma^2} := \{\sigma_1^2, \cdots, \sigma_K^2\}$ for simplicity. The following theorems give the bounds for the case of $p = \infty$ and $p < \infty$ respectively.

**Theorem 3.1.** *For $p = \infty$, we have*

$$\mathbb{E}\left[R_p(\boldsymbol{n}_{\pi_1}) - R_p(\boldsymbol{n}^*)\right] \leq 4\sqrt{2}\sigma^2 \mathcal{F}_{\text{Alg 1},\infty}(\lambda, \boldsymbol{\sigma^2})$$
$$\cdot T^{-3/2}\sqrt{\log T} + o(T^{-3/2}),$$

*where $\mathcal{F}_{\text{Alg 1},\infty}(\lambda, \boldsymbol{\sigma^2}) := \lambda^{-1/2}(K + \Sigma_2/\underline{\sigma}^2 - 2)$.*

**Theorem 3.2.** *For $p < \infty$, we have:*

$$\mathbb{E}\left[R_p(\boldsymbol{n}_{\pi_1}) - R_p(\boldsymbol{n}^*)\right] \leq 24\sigma^4 \mathcal{F}_{\text{Alg 1},p}(\lambda, \boldsymbol{\sigma^2})$$
$$\cdot T^{-2} \log T + o(T^{-2}),$$

*where $\mathcal{F}_{\text{Alg 1},p}(\lambda, \boldsymbol{\sigma^2}) := \frac{p^2(\Sigma_q)^{1/p}\Sigma_{q-4}}{\lambda(p+1)}$.*

The analyses for the case of infinite and finite $p$ are largely similar, with minor differences caused by the optimality structure of $\boldsymbol{n}^*$. For both cases, we note that the optimal value $R_p(\boldsymbol{n}^*)$ is on the order of $1/T$; therefore, the excess error of the algorithm is of lower order than the oracle value, with optimality gaps of order $T^{-3/2}\sqrt{\log T}$ for $p = \infty$ and $T^{-2} \log T$ for finite $p$. These rates are also essentially tight in their polynomial dependence on $T$. In particular, Aznag et al. (2023) establish lower bounds via Le Cam's two-point method in the Gaussian setting, showing that the optimality gap is at least $\Omega(T^{-3/2})$ for $p = \infty$ and $\Omega(T^{-2})$ for finite $p$. Since the two-arm Gaussian instance is a special case of the general multi-group mean estimation problem, these lower bounds certify that our upper bounds match the known minimax exponents up to logarithmic factors.

We make the following remarks about the results. First, the bounds improve on orders of $\log T$ compared to the respective results in Aznag et al. (2023). The main differences of our algorithm and analysis are (i) the refined variance concentration inequality in Lemma 2.4 and (ii) getting rid of the UCB design. This reinforces our point that solving the problem of multi-group mean estimation can be exploration-free. Moreover, the design of Algorithm 1 is closely related to the Neyman allocation principle (Zhao, 2024). With respect to comparisons to adaptive Neyman allocation methods, we note that the existing literature predominantly focuses on the special case of $K = 2$ arms with a $p = 1$ objective. While these algorithms could be extended to general $K$ and $p$, the restriction to $K = 2$ substantially simplifies the problem structure and the associated theoretical analysis. In addition, we emphasize that the bounds can be further refined under the case of strictly subgaussian or gaussian, which we defer to Appendix C.3.

## 4. General Case

In the previous section, we consider the case where a lower bound $\underline{\sigma}^2$ for the variances is known a priori. Now we consider the general case where there is no such prior knowledge. We note that in Algorithm 1 the only point where we use $\underline{\sigma}^2$ is to determine the length of the initial phase $\tau$. The knowledge of $\underline{\sigma}^2$ ensures that we will not exhaust the optimal budget $n_k^*$ in the initial phase. Thus, the idea of our second algorithm is to replace the knowledge of $\underline{\sigma}^2$ with some variance estimate based on the collected observations. Accordingly, the allocation scheme will be more adaptively adjusted based on the data flow.

To simplify the notations, suppose some LCB and UCB estimates for the variances satisfy

$$\mathbb{P}\left(\text{LCB}_{k,n} \leq \sigma_k^2 \leq \text{UCB}_{k,n} \text{ for all } k \text{ and } n\right) \geq 1 - 2T^{-c}$$

where $c$ will be determined by $p$. Here the event is taken as a union over all the alternatives $k$ and all the number of observations $n$ (up to $T$). Specifically, for each alternative $k$, one can construct such confidence bounds of $\text{LCB}_{k,n}$ and $\text{UCB}_{k,n}$ based on the sample variance estimator and Lemma 2.4. The width of the confidence interval can be adjusted to ensure that the *good* event (of true variances falling in confidence bounds uniformly) happens with a high probability. The detailed components of the LCBs and UCBs and the value of $c$ are deferred to Appendix C.2.

---

**Algorithm 2** Adaptive Algorithm

---

1: **Input:** Time horizon $T$, constant $q$, constant $m$.
2: **Phase 1: Avoid a meaningless LCB**
3: Select each alternative $n = O(1)$ times such that

$$\min_{k=1,\ldots,K} \text{LCB}_{k,n} > 0$$

4: **Phase 2: Determine stopping times**
5: Initialize $\mathcal{A}_{\text{active}} \leftarrow \{1, \ldots, K\}$
6: Initialize $n_k \leftarrow n$, $\lambda_{k,n_k} \leftarrow n/T$.
7: **repeat**
8:     **for** each $k \in \mathcal{A}_{\text{active}}$ **do**
9:         Select $k$ for $\lambda_{k,n_k}T - n_k$ times
10:         Update $n_k \leftarrow \lambda_{k,n_k}T$
11:     **end for**
12:     Compute $\text{LCB}_{k,n_k}, \text{UCB}_{k,n_k}$ for all $k$.
13:     **for** each $k = 1, \ldots, K$ **do**
14:         Update $\lambda_{k,n_k} = \frac{\text{LCB}_{k,n_k}^{q/2}}{\text{LCB}_{k,n_k}^{q/2} + \sum_{j \neq k} \text{UCB}_{j,n_j}^{q/2}}$
15:         **if** $n_k \geq \lambda_{k,n_k}T$ **then**
16:             Set $\tau_k \leftarrow n_k$, and remove $k$ from $\mathcal{A}_{\text{active}}$
17:         **else**
18:             Add $k$ back to $\mathcal{A}_{\text{active}}$ if $k \notin \mathcal{A}_{\text{active}}$
19:         **end if**
20:     **end for**
21: **until** $\mathcal{A}_{\text{active}} = \emptyset$
22: **Phase 3: Allocate the remaining periods**
23: **for** each $k = 1, \cdots, K$ **do**
24:     Compute $\hat{\sigma}^2_{k,\tau_k}$ and calculate $\lambda_{k,\tau_k}$
25:     Select $k$ for $\lambda_{k,\tau_k}T - \tau_k$ rounds
26:     Calculate $\hat{\mu}_k(T) = \frac{1}{\lambda_{k,\tau_k}T} \sum_{s=1}^{\lambda_{k,\tau_k}T} X_{k,s}$
27: **end for**
28: **Output:** Final estimates $\{\hat{\mu}_k(T)\}_{k=1}^K$.

---

Algorithm 2 presents our adaptive algorithm for the general case, i.e., without the knowledge of $\underline{\sigma}^2$. Phase 1 of the algorithm is a trivial part that simply aims to ensure all the LCB estimates are positive. The key of Phase 2 is the quantity $\lambda_{k,n_k}$, which implies a lower bound for $n_k^*$. The active set $\mathcal{A}_{\text{active}}$ maintains the alternatives that still need some more rounds of selection towards the optimal allocation. The quantity $\lambda_{k,n_k}$ is closely related the constant

$\lambda$ and the initial length $\tau$ in Algorithm 1 where we replace LCBs and UCBs with $\underline{\sigma}^2$ and $\sigma^2$. While both designs aim to ensure that we don't over-select an alternative, LCBs and UCBs are more adaptive to the data, and thus Algorithm 2 should give a better performance. The last phase of the algorithm simply exhausts the remaining time steps as the second phase of Algorithm 1.

In the literature of multi-armed bandits and best-arm identification, there is a line of works (Auer & Ortner, 2010; Karnin et al., 2013; Soare et al., 2014; Qian & Yang, 2016) that utilize adaptive arm elimination (as Phase 2 of Algorithm 2). Among this stream of works and algorithms, our arm elimination procedure is most similar to Procedure 1 in Cai et al. (2024) which focuses on the reward objective, and the algorithm in Li et al. (2021) which also incorporates LCBs/UCBs into the optimization problem to perform arm elimination.

**Theorem 4.1.** *We have the following results for Algorithm 2. For $p = \infty$,*

$$\mathbb{E}\left[R_p(\boldsymbol{n}_{\pi_2}) - R_p(\boldsymbol{n}^*)\right] \leq 8\sigma^2 \mathcal{F}_{\text{Alg }2,\infty}(\boldsymbol{\sigma^2}) \\ \cdot T^{-3/2}\sqrt{\log T} + o(T^{-3/2}),$$

*and for $p$ is finite, we have:*

$$\mathbb{E}\left[R_p(\boldsymbol{n}_{\pi_2}) - R_p(\boldsymbol{n}^*)\right] \leq 40\sigma^4 \mathcal{F}_{\text{Alg }2,p}(\boldsymbol{\sigma^2}) \\ \cdot T^{-2}\log T + o(T^{-2}),$$

*where $\mathcal{F}_{\text{Alg }2,p}(\boldsymbol{\sigma^2}) := \frac{p^2(\Sigma_q)^{2/q}(\Sigma_{-4})}{p+1}$, $\mathcal{F}_{\text{Alg }2,\infty}(\boldsymbol{\sigma^2}) := \sqrt{\Sigma_2}\left(\Sigma_{-1} + \frac{\Sigma_2}{\sigma_{\min}^3} - \frac{2}{\sigma_{\min}}\right)$, and $\boldsymbol{n}_{\pi_2}$ denotes the allocation scheme of Algorithm 2.*

Theorem 4.1 gives the performance bounds for Algorithm 2. The analysis shares a similar spirit with that of Algorithm 1 but it deals with some additional complications caused by the LCBs and UCBs. We make several remarks for Algorithm 2 and Theorem 4.1. First, as in the case of Algorithm 1, though Algorithm 2 involves the elements of LCBs and UCBs, it is still exploration-free. In other words, LCBs and UCBs arise from inaccurate estimates of the variance, but they don't incur any redundant play of any alternative $k$. Second, Algorithm 2 gives a bound on the same order as Algorithm 1 under the scenario of no prior knowledge on $\underline{\sigma}^2$. Essentially, one can establish (from the analyses of these two algorithms) that as long as (i) each alternative is played $\Omega(T)$ (ii) the number of plays doesn't exceed the optimal scheme $n_k^*$, then one can always achieve an optimal gap as the ones in Theorems 3.1, 3.2, and 4.1. Either the knowledge of $\underline{\sigma}^2$ or the LCB/UCB design in Algorithm 2 is used to ensure these two conditions. Lastly, we note that a strictly subgaussian distribution will give better bounds both theoretically and numerically. In particular, it eliminates the need for prior knowledge of $\sigma^2$ in constructing LCBs and

UCBs, and further reduces the sampling requirement in the first phase of Algorithm 2 to a small constant that depends only on $T$ and $c$. We defer to Appendix C.3 for a detailed discussion. Besides, we give a theoretical comparison between ours and the previous result in Aznag et al. (2023) in Appendix C.5.

# 5. Contextual Case

In this section, we extend the multi-group estimation to a contextual bandits setting where the goal is to estimate the group-level linear parameters rather than just the means of rewards. We adopt the multiple linear contextual bandit model described in Slivkins (2011; 2019) and follow the notations in the previous sections to ensure consistency throughout the paper.

## 5.1. Problem Setting

Consider a contextual bandit setting with $K$ arms, where each arm $k \in \{1, \ldots, K\}$ is associated with an unknown parameter vector $\beta_k \in \mathbb{R}^d$. At each round $t = 1, \ldots, T$ (assuming $d \ll T$), a context vector $c_t \in \mathbb{R}^d$ is observed, drawn i.i.d. from a distribution $\mathcal{P}_C$. In other words, we consider the setting of *stochastic context*. For $\mathcal{P}_C$, we make the following assumptions.

**Assumption 5.1.** We assume $\Sigma := \mathbb{E}_{c \sim \mathcal{P}_C}\left[cc^\top\right] \succ 0$. Moreover, $\|c_n\| \leq R < \infty$ almost surely.

Let $\lambda_{\min}^C := \lambda_{\min}(\Sigma)$ and $\lambda_{\max}^C := \lambda_{\max}(\Sigma)$ represent the minimum eigenvalue respectively. Upon pulling arm $k$, the observed reward is given by

$$X_{k,n} = \beta_k^\top c_n + \eta_{k,n},$$

where $\eta_{k,n}$ is zero-mean noise, i.i.d. across time and arms, but with unknown arm-dependent variances $\sigma_k^2$. And we assume that $\eta_{k,n}$'s are subgaussian and satisfy Assumption 2.3. The goal is to determine the number of times needed for each arm beforehand to estimate the parameter vector $\beta_k$ for each arm, and to evaluate the overall estimation error using the squared $\ell_2$-norm:

$$\min \ \mathbb{E}_{\mathcal{C},\eta}\left[\sum_{k=1}^{K}\|\hat{\beta}_{k,n_k} - \beta_k\|^2\right] \tag{6}$$

$$\text{s.t.} \quad \sum_{k=1}^{K} n_k = T.$$

where $\hat{\beta}_{k,n} = V_{k,n}^{-1}\sum_{s=1}^{n} c_{k,s}X_{k,s}^\top$, and $V_{k,n} = \gamma \cdot I_d + \sum_{s=1}^{n} c_{k,s}c_{k,s}^\top$ with $\gamma$ is the (ridge) penalty factor and $I_d$ is $d$-dimensional identity matrix. We remark that Riquelme et al. (2017) and Fontaine et al. (2021) consider a similar objective to ours. The optimization problem above can be viewed as a natural extension of that for the multi-group mean estimation.

## 5.2. Algorithm and Analysis

First, the following lemma characterizes the estimation error. In contrast to the setting in the previous sections, the error involves the number of observations $n$ in a much more complicated manner. Specifically, we note that the inverse sample covariance matrix appears in the expression. This prevents the usage of UCB-type algorithms (such as Aznag et al. (2023)) for this contextual multi-group estimation; this is because a UCB-based algorithm can provide an error bound in the data-dependent norm $\|\cdot\|_{V_{k,n}^{-1}}$ but not in the Euclidean norm $\|\cdot\|$. The data-dependent norm suffices for deriving a standard regret bound with the help of the elliptical potential lemma but cannot be transformed to a bound for (6).

**Lemma 5.2.** *Let $C_{k,n} = [c_{k,1}, \cdots, c_{k,n}]$ be the first $n$ context vectors shown for arm $k$, then we have:*

$$\mathbb{E}\left[\|\hat{\beta}_{k,n} - \beta_k\|^2 \mid C_{k,n}\right] = \sigma_k^2 \operatorname{Tr}(V_{k,n}^{-1}) + \gamma^2 \beta_k^\top V_{k,n}^{-2}\beta_k$$
$$- \gamma\sigma_k^2 \operatorname{Tr}(V_{k,n}^{-2})$$

*where the expectation is taken w.r.t. the noises $\eta_{k,n}$'s.*

In our algorithm problem, we use the ridge regression to estimate $\beta_k$'s as the design in the linear bandits literature. This prevents the singularity of the sample covariance matrix. Before we proceed, we first present the matrix concentration inequality.

**Theorem 5.3.** *(Theorem 6.1.1 in Tropp (2015)) Let $\{X_i\}_{i=1}^{n}$ be independent, centered, self–adjoint random matrices in $\mathbb{R}^{d \times d}$ with $\mathbb{E}[X_i] = 0$ and $\|X_i\| \leq R$ a.s., let $\nu = \left\|\sum_{i=1}^{n} \mathbb{E}[X_i^2]\right\|$, then for every $t \geq 0$:*

$$\mathbb{P}\left\{\left\|\sum_{i=1}^{n} X_i\right\| \geq t\right\} \leq 2d \cdot \exp\left(-\frac{t^2/2}{\nu + Rt/3}\right).$$

As an implication, we can derive the following bound for the ridge regression estimator.

**Lemma 5.4.** *Let $\gamma = \lambda_{\min}^C/n$, and for $n \geq 2$, we have:*

$$\mathbb{E}\left[\|\hat{\beta}_{k,n} - \beta_k\|^2\right] \leq \frac{2d\sigma_k^2}{n\lambda_{\min}^C} + o(n^{-2}).$$

From the lemma, we can approximately represent the objective (6) with the following optimization problem. The rationale is that the difference between the following problem and (6) is of a lower order. We point out that this approximation requires an additional condition for the algorithm design, which we will address after describing the

algorithm.

$$\min \quad R_T(\boldsymbol{n}) = \frac{2d}{\lambda_{\min}^{\mathcal{C}}} \sum_{k=1}^{K} \frac{\sigma_k^2}{n_k}$$

$$\text{s.t.} \quad \sum_{k=1}^{K} n_k = T$$

where the decision variables are the allocation scheme $\boldsymbol{n} = (n_1, ..., n_K)$. The optimal solution of this problem is $n_k^* = \frac{\sigma_k}{\Sigma_1} \cdot T$. Now the problem reduces to estimating $\sigma_k^2$, which is similar to the group-mean estimation setup in the previous sections. Then we can define the residual term as:

$$r_{k,s} = X_{k,s} - \hat{\beta}_{k,n}^{\top} c_{k,s}$$

and the estimated variance becomes:

$$\hat{\sigma}_{k,n}^2 = \frac{1}{n-1} \sum_{s=1}^{n} (r_{k,s} - \frac{1}{n} \sum_{s=1}^{n} r_{k,s})^2 \qquad (7)$$

---

**Algorithm 3** Contextual Algorithm

1: **Input:** Time horizon $T$, context distribution $\mathcal{P}_{\mathcal{C}}$, context dimension $d$, minimum eigenvalue $\lambda_{\min}^{\mathcal{C}}$.
2: Play each arm $d$ times $\{X_{k,n}\}_{n=1}^{d}$ and $\{c_{k,n}\}_{n=1}^{d}$
3: **Phase 1: Avoid a meaningless LCB**
4: Select each alternative $n = O(1)$ times such that

$$\min_{k=1,...,K} \text{LCB}_{k,n} > 0$$

5: **Phase 2: Adaptive Elimination Strategy**
6: Employ the second phase of Algorithm 2 until $\mathcal{A}_{\text{active}} = \emptyset$ based on $\hat{\sigma}_{k,n}^2$ with (7).
7: **Phase 3: Allocate the remaining periods**
8: **for** each arm $k = 1, \cdots, K$ **do**
9:    Compute $\hat{\sigma}_{k,n_k}^2$ and $\lambda_{k,n_k}$
10:    Play arm $k$ for $\lambda_{k,n_k} \cdot T - n_k$ rounds
11:    Calculate $\hat{\beta}_k(T) = V_{k,n_k}^{-1} \sum_{s=1}^{n_k} c_{k,s} X_{k,s}^{\top}$
12: **end for**
13: **Output:** Final estimates $\{\hat{\beta}_k(T)\}_{k=1}^{K}$

---

The algorithm follows an almost identical structure with Algorithm 2 by replacing the variance estimates by (7). An important design of the algorithm is that in Phase 2, the allocation decision (which arm to select at time $t$) is decided before seeing the context $c_t$. This design is quite different from other algorithms on linear bandits. On one hand, this is admitted by the nature of the group-mean estimation, which requires each arm to be played for $\Omega(T)$ times. On the other hand, this first-decide-then-observe structure ensures the independence between context vectors shown for each arm, and hence makes Theorem 5.3 and Lemma 5.4 applicable.

The following theorem gives the performance bound for Algorithm 3, which is comparable to the finite-$p$ case in the previous sections.

**Theorem 5.5.** *For Algorithm 3, with $p = 1$, we have*

$$R_T(\boldsymbol{n}_{\pi_3}) - R_T(\boldsymbol{n}^*) \leq \frac{80d\sigma^2}{\lambda_{\min}^{\mathcal{C}}} \mathcal{F}_{\text{Alg } 3,p}(\boldsymbol{\sigma^2})$$
$$\cdot T^{-2} \log T + o(T^{-2}),$$

*where $\boldsymbol{n}_{\pi_3}$ denotes the allocation scheme of Algorithm 3, and $\mathcal{F}_{\text{Alg } 3,p}(\boldsymbol{\sigma^2}) = \mathcal{F}_{\text{Alg } 2,p}(\boldsymbol{\sigma^2})$.*

## 6. Numerical Experiments

In this section, we present numerical experiments for our algorithms where all the results are reported based on 100 simulation trials. We also refer to Appendix D for more experiments and details.

### 6.1. Gaussian alternatives

For traditional bandit problems, we play $K = 4$ arms generated from Gaussian distribution $\mathcal{G}_k$ with mean $\mu_k \sim \mathcal{U}([-1, 1])$ and $\{\sigma_1^2, \sigma_2^2, \sigma_3^2, \sigma_4^2\} = \{1, 1.5, 2, 2.5\}$ respectively. In this setting, we have $\sigma^2 = 2.5$ and $\underline{\sigma}^2 = 1$ if known. We conduct the experiment under the general subgaussian (GSG) setting and the strictly subgaussian (SSG) setting, respectively. From Figure 1, we observe a sharp performance drop when $\underline{\sigma}^2$ is unknown. Moreover, the point at which this drop occurs differs between the GSG and SSG settings. This discrepancy is caused by different lengths of Phase 1 in Algorithm 2. In the GSG setting, this length necessitates a large time horizon $T$, more than $2 \times 10^4$. When $T$ is not large enough, the effective exploration budget per arm is only about $T/K$. Besides, we note that the theoretical upper bound in the SSG setting is much closer to the empirical regret, particularly when $p = \infty$, which corroborates the sharper guarantees predicted by our analysis. More visualizations are deferred to Appendix D.2.

### 6.2. Rademacher and Gaussian alternatives

Now we reproduce the numerical experiment of Non-Gaussian arms in Carpentier et al. (2011). This is a two-arm bandit problem: one with a Gaussian $\mathcal{N}(0, \sigma_1^2)$ with $\sigma_1^2 \geq 1$, and another with Rademacher arm. They used $\sigma^2 = \sigma_1^2 + 1$ as the prior information and $\lambda_{\min} = 1/(1 + \sigma_1^2)$. In their experiment, they showed the $p$-norm result with $p = \infty$ of $T = 10^3$ for different $\sigma_1^2$. In the GSG setting, the length of the first phase of Algorithm 2 is related to $\sigma^2$. For a large $\sigma_1^2$ a sufficiently large horizon $T$ is required to realize the theoretical guarantees. By contrast, in the SSG setting the first-phase length reduces to a constant independent of $\sigma$, leading to a faster and simpler procedure. Since both the Gaussian and Rademacher distributions belong to the

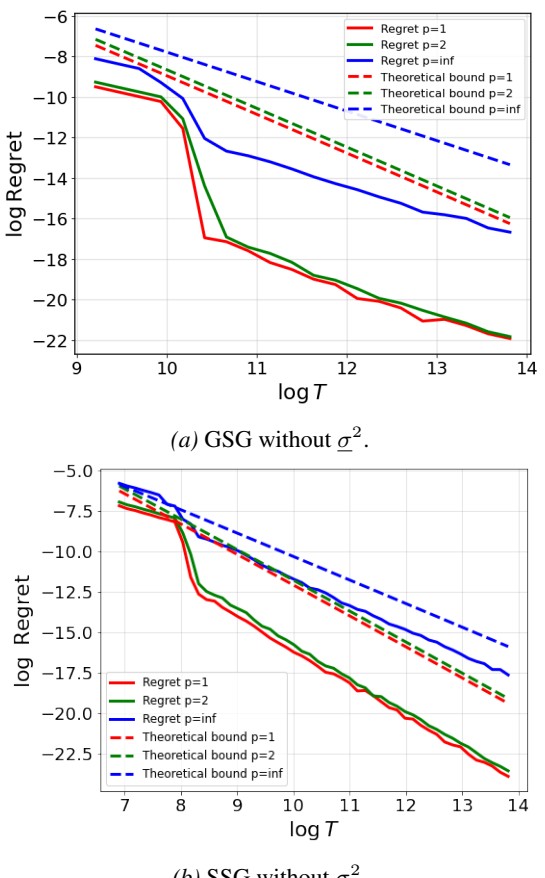

*(a)* GSG without $\underline{\sigma}^2$.

*(b)* SSG without $\underline{\sigma}^2$.

*Figure 1.* Algorithm 2: Gaussian alternatives under two settings.

class of strictly subgaussian distributions, we conduct experiments in the SSG setting with $\sigma_1^2 \in \{5, 20, 50, 100\}$. Figure 2 shows the empirical regret (solid line) compared against the theoretical upper bound (dashed line) for different $\sigma_1^2$.

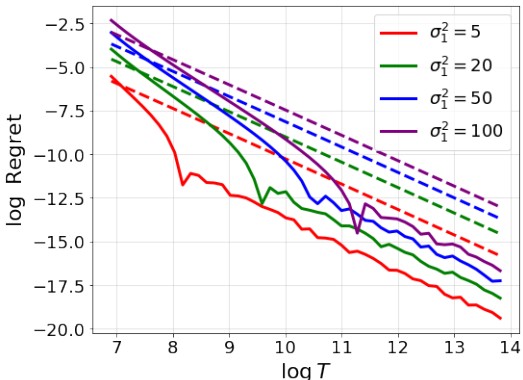

*Figure 2.* Algorithm 2: Rademacher and Gaussian alternatives in SSG setting.

### 6.3. Contextual Bandit

We consider a contextual bandit setting with varying numbers of arms $K \in \{5, 10, 20\}$ and dimension $d = 4$. Context vectors are sampled uniformly from the hypercube $\left[-\sqrt{3}, \sqrt{3}\right]^d$. For each arm $k$, the coefficient vector $\beta_k$ is independently drawn from $\mathcal{U}[-2, 2]^d$, while the noise $\eta_k$ follows a Gaussian distribution with variance sampled from $\mathcal{U}[1, 4]$. We assume $\underline{\sigma}^2 = 1$ and $\sigma^2 = 4$, and evaluate the performance under both the GSG and SSG settings. As shown in Figure 3, the empirical regret exhibits a slope close to $-2$, which is consistent with the rate predicted by Theorem 5.5.

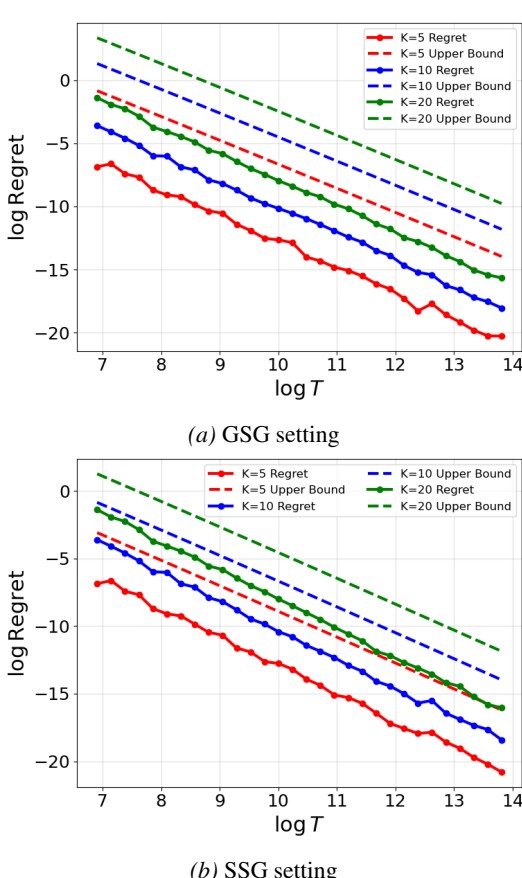

*(a)* GSG setting

*(b)* SSG setting

*Figure 3.* Algorithm 3: Contextual setting.

## 7. Conclusion and Discussions

In this paper, we study the multi-group mean estimation problem and consider both the canonical and the contextual settings of the problem. We propose several algorithm that features a simpler design than the existing ones but achieves the optimal order of regret. The proposed algorithms and analyses reveal several key structural insights to the problem: First, the optimal allocation scheme requires $\Omega(T)$ selections of each alternative/arm, and thus it

enables exploration-free algorithms. Second, we point out a connection between the problem and the best arm identification problem, where the optimal allocation scheme is not in closed form. Third, we explain why the UCB-type exploration is unnecessary for the canonical setting and how it completely fails in the contextual setting. As a side product, we find that a strictly subgaussian distribution allows a sharper theoretical bound and also a better numerical performance. It suggests a potential research direction: If the reward distributions $\mathcal{P}_k$ belong to such structured families, but the variances are only partially available, one could develop adaptive algorithms that exploit this structure. Such algorithms could iteratively refine their variance estimates and thereby achieve tighter confidence intervals, leading to further improved sample efficiency.

## Impact Statement

This paper presents work whose primary goal is to advance the theory and methodology of machine learning, particularly for multi-group mean estimation and adaptive allocation under limited sampling budgets. The proposed algorithms are designed to improve the statistical efficiency of estimating multiple group-level quantities simultaneously, rather than concentrating samples only on the most rewarding or most favorable group. As such, this work may have positive societal impact in applications where reliable estimates across multiple populations are important, including survey design, experimental design, adaptive A/B testing, personalized decision-making, and policy evaluation.

Our results are theoretical and rely on statistical assumptions such as subgaussian noise and, in the contextual setting, stochastic context distributions. If these assumptions are violated in practice, the resulting confidence estimates and allocation decisions may be unreliable. We therefore view the methods developed here as tools for improving sample-efficient statistical estimation, rather than as fully automated decision-making systems. Responsible use requires validating the modeling assumptions, monitoring empirical performance across groups, and incorporating ethical, legal, and institutional constraints appropriate to the application domain.

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

# A. Related Works

In this section, we discuss more related works to our paper.

**Variance concentration.**    Variance concentration is fundamental for assessing the reliability of empirical estimates. Classical Bernstein inequalities yield variance-sensitive bounds that improve over Hoeffding-type results. Maurer & Pontil (2009) introduced empirical Bernstein bounds that replace the true variance with its empirical estimate under boundedness assumptions. Carpentier et al. (2011) extended this to unbounded distributions via a truncation-based technique. Recently, Martinez-Taboada & Ramdas (2025) provided sharper empirical Bernstein bounds for bounded case based on a time-uniform Chernoff bound in Howard et al. (2020). Further refinements arise from the Hanson-Wright inequality, which provides sharp tail bounds for quadratic forms of subgaussian vectors and underpins concentration results for covariance and design matrices. Rudelson & Vershynin (2013) gave a modern proof with unspecified constants. For the notion of subgamma, Laurent & Massart (2000) gave a proof for Gaussian distribution, and Boucheron et al. (2013) followed the proof framework and introduced the notion of subgamma, based on which Epperly (2022) gave a proof of Hanson-Wright inequality with specific constants. Besides subgamma, another closely related and widely-used notion is subexponential distribution. Vershynin (2018), Wainwright (2019), Rigollet & Hütter (2023) gave detailed introduction of subexponential distribution.

**Group mean estimation in MAB.**    Group mean estimation is an important problem in the MAB framework, as many applications require uniformly accurate estimates of all arm means rather than simply identifying the best arm. Shin et al. (2019a;b) analyzed the statistical properties of sample means under adaptive sampling. In this setting, the goal is to allocate sampling resources strategically to minimize estimation error under limited feedback. A major line of research builds on UCB-based algorithms, which is based on Lai & Robbins (1985) and Auer et al. (2002). Antos et al. (2008; 2010) were among the first to formally study active learning in MABs with heteroscedastic noise. Their work analyzed optimal allocation strategies for minimizing estimation error under bounded reward assumptions. Carpentier et al. (2011) mentioned this topic is related to pure-exploration and provided new regret bounds by proposing UCB-based algorithms under unbounded case. Recently, Aznag et al. (2023) revisited this classical allocation problem using a $p$-norm objective to capture different notions of group-level estimation quality, and derived the theoretical lower bound. Besides estimating the means of distributions in squared error sense, Shekhar et al. (2020) considered four general distance measures: $\ell_2^2$, $\ell_1$, $f$-divergence, and separation distance. For linear models, especially linear bandit and contextual bandit, the estimation becomes to estimate the linear coefficient $\beta$. Riquelme et al. (2017) studied this problem setting in the multi-linear regression with the assumption that both noise and context are generated from Gaussian distribution. When the mean of arms is a linear combination with an unknown parameter, then the problem becomes an optimal experimental design problem (Pukelsheim, 2006; Sabato & Munos, 2014; Dimakopoulou et al., 2019; Allen-Zhu et al., 2021; Khamaru et al., 2021; Fontaine et al., 2021; Simchi-Levi & Wang, 2025). Specifically, Fontaine et al. (2021) considered this univariate unknown parameter under heterogeneous subgaussian noise with unit-norm context vectors.

**Neyman Allocation.**    Neyman allocation (Neyman, 1934) has become a foundational principle in experimental design for group mean and average treatment effect estimation. In the classical setting where group variances are known, sampling according to the Neyman rule minimizes the asymptotic variance of the difference-in-means estimator and thus serves as a natural benchmark for efficient estimation. In practical applications such as A/B testing and causal inference, however, group variances are typically unknown, which motivates adaptive allocation strategies that balance exploration for variance estimation with exploitation for statistical efficiency. Zhao (2024) formalize this trade-off in a multi-stage adaptive Neyman allocation framework and propose an algorithm that approximates the oracle Neyman allocation. Related lines of work have also studied Neyman-type allocation in fully online or sequential settings, analyzing performance guarantees under continuous adaptation (Dai et al., 2023; Noarov et al., 2025).

**Exploration-free algorithms.**    Lattimore & Szepesvári (2020) gave an introduction of pure exploration. For exploration-free algorithms, one of the important techniques in the exploration period is arm elimination, which is widely used as a core method for best arm identification (Auer & Ortner, 2010; Audibert & Bubeck, 2010; Karnin et al., 2013; Soare et al., 2014; Qian & Yang, 2016). In the contextual bandit setting, Bastani et al. (2021) introduced the notion of natural exploration and showed that a warm-start greedy policy can be near-optimal with little explicit exploration. Hao et al. (2020) developed an optimization-based adaptive exploration scheme that tracks the instance-optimal sampling allocation and achieved instance-dependent asymptotic optimality, with sub-logarithmic regret under rich context distributions, spurring a growing line of follow-up work over the recent years. Wan et al. (2022) studied safe exploration for policy evaluation and

comparison, formulating data collection as a constrained design problem and deriving exploration policies that ensure safety while improving the statistical efficiency of off-policy evaluation.

**Conservative bandits.** Conservative bandits study safety-constrained exploration that keeps performance close to a baseline while learning. Unlike exploration-free algorithms, conservative methods do not eliminate exploration, they restrict it to the minimum required to satisfy safety, so under strong baselines or tight constraints they may appear nearly greedy while still performing essential, safety-driven probing (Wu et al., 2016; Kazerouni et al., 2017; Amani et al., 2019; Garcelon et al., 2020).

## B. Concentration Inequality for Subgaussian Variance Estimation

Accurate variance estimation plays a central role in the design and analysis of the algorithms developed in this paper, particularly in the context of active learning and adaptive allocation strategies. This section presents a collection of theoretical results that characterize the concentration behavior of empirical variance estimators, we review and extend several classical results under different distribution assumptions.

### B.1. Preliminaries and Key Definitions

In the context of variance estimation, it is important to note that both the sample mean and the sample variance can be formulated as U-statistics. A key theoretical property of U-statistics is that they satisfy a central limit theorem, which ensures their asymptotic normality under mild regularity conditions (Van der Vaart, 2000).

From Definition 2.2 of subgaussian, it is straightforward to verify that $\mathbb{E}[X] = 0$ and the variance $\sigma_X^2 \leq \sigma^2$. A particularly interesting case arises when equality holds, i.e., $\sigma^2 = \sigma_X^2$. This condition is satisfied by a special class of distributions named strictly subgaussian distribution (Arbel et al., 2020; Bobkov et al., 2024). It includes examples such as Gaussian, Rademacher, symmetric Beta distribution and so on.

While for some specific distribution types, symmetric distributions is crucial for strictly-subgaussian, but generally, symmetry is neither necessary nor sufficient for strictly-subgaussian, see Proposition 1.1 and 1.2 in Arbel et al. (2020). When $X$ is strictly-subgaussian with variance $\sigma_X^2$, we could get that $\mathbb{E}[X^3] = 0$ and $\mathbb{E}[X^4] \leq 3\sigma_X^4$ by using Taylor expansion in Equation (2). We adopt the notion of subgamma introduced in Section 2.4 of Boucheron et al. (2013) as a key tool for establishing variance concentration:

**Definition B.1.** A real-valued centered random variable $X$ is said to be $(\nu, c)$-subgamma on the right tail with variance factor $\nu$ and scale parameter $c$, denoted as $\Gamma_+(\nu, c)$, if for every $t$ such that $0 < t < 1/c$:

$$\psi_X(t) = \log \mathbb{E}[\exp(tX)] \leq \frac{t^2 \nu}{2(1 - ct)},$$

where $\psi_X(t)$ is the cumulant generating function. Similarly, $X$ is said to be $(\nu, c)$-subgamma on the left tail, denoted as $\Gamma_-(\nu, c)$, if $-X$ belongs to $\Gamma_+(\nu, c)$. If $X$ is $(\nu, c)$-subgamma on both tails, then such random variable is denoted as $\Gamma(\nu, c)$.

Apart from the subgamma distribution, another closely related and widely used notion is that of subexponential distributions. The theoretical results associated with these two distributional assumptions are largely interchangeable, as many concentration inequalities derived under one setting can be reformulated under the other with comparable bounds. The main distinction lies in the definition: subexponential is typically defined symmetrically for two-sided tails, whereas subgamma treats tails separately. The relationship between subgaussian and subgamma is as follows:

**Lemma B.2.** *If $X$ follows $\sigma$-subgaussian distribution, then $X^2 - \mathbb{E}[X^2]$ belongs to $\Gamma_+(16\sigma^4, 2\sigma^2)$ and $\Gamma_-(16\sigma^4, \sigma^2/3)$. Specifically, if $X$ follows strictly subgaussian, then $X^2 - \mathbb{E}[X^2]$ belongs to $\Gamma_+(2\sigma_X^4, 2\sigma_X^2)$ and $\Gamma_-(2\sigma_X^4, \sigma_X^2/3)$.*

*Remark* B.3. For Lemma B.2, if extra information of $X$ is available, such as symmetry or bounded, then we can get more accurate value of $\nu$ and $c$. For example, if $X$ is both symmetric and strictly-subgaussian, then for all $t \in \mathbb{R}$:

$$\psi_{X^2 - \mathbb{E}[X^2]}(t) \leq \frac{\sigma_X^4 t^2}{1 - 2\sigma_X^2 t},$$

which means $X^2 - \mathbb{E}[X^2]$ belongs to $\Gamma_+(2\sigma_X^4, 2\sigma_X^2)$ and $\Gamma_-(2\sigma_X^4, 0)$. One typical case is $X \sim \mathcal{N}(0, \sigma_X^2)$.

## B.2. Subgaussian Variance Concentration

We first revisit several commonly used concentration inequalities relevant to the sample variance. For clarity, let the sample variance of $\{X_1, \cdots, X_n\}$ is defined as:

$$\hat{\sigma}_n^2 = \frac{1}{n-1} \sum_{i=1}^{n} (X_i - \bar{X})^2 \tag{8}$$

We assume $\mu_X = 0$ for simplicity, in the case where $\mu_X \neq 0$, the expression can be equivalently rewritten by replacing $X_i$ with $X_i - \mu_X$ in Equation (8). For $X$ is bounded random variable, one of the famous concentration inequality is Theorem B.4:

**Theorem B.4.** *(Theorem 10 in Maurer & Pontil (2009)) Let $\{X_1, \cdots, X_n\}$ be $n \geq 2$ i.i.d. random variables with variance $\sigma_X^2$ and such that $\{X_i\}_{i=1}^{n} \in [0, b]$. Then with probability at least $1 - 2\delta$, we have:*

$$|\hat{\sigma}_n - \sigma_X| \leq b\sqrt{\frac{2\log(1/\delta)}{n-1}}.$$

For Theorem B.4, Martinez-Taboada & Ramdas (2025) recently established a sharper empirical Bernstein inequality for bounded observations $X_i \in [0, 1]$, building on the time-uniform Chernoff bound of Howard et al. (2020). However, extending such results beyond the bounded regime remains challenging. Among the few existing works addressing unbounded random variables, Carpentier et al. (2011) provides a representative concentration bound, which has been widely adopted in subsequent studies (Aznag et al., 2023).

**Theorem B.5.** *(Lemma 4 in Carpentier et al. (2011), rephrased) Let $X$ follows $\sigma$-subgaussian and variance $\sigma_X^2$, then with probability at least $1 - 2\delta$, we have:*

$$|\hat{\sigma}_n - \sigma_X| \leq 2\sigma\sqrt{2\log(1/\delta)}\sqrt{\frac{2\log(2/\delta)}{n-1}} + \frac{2\sigma\sqrt{\delta(2 - \log(\delta))}}{1-\delta}. \tag{9}$$

Theorem B.5 is an extension of Theorem B.4 under unbounded case, derived through a truncation-based technique at $|X| \leq \sqrt{2\sigma^2 \log(1/\delta)}$. However, it could be optimized by utilizing the notion of subgamma and Hanson-Wright inequality. This inequality is particularly powerful in controlling the tail behavior of second-order chaos variables, and is widely applied in analyzing the concentration of quadratic forms.

**Theorem B.6.** *(Theorem 6.2.1 in Vershynin (2018), Epperly (2022)) Let $\boldsymbol{X} \in \mathbb{R}^n$ have independent, mean-zero, $\sigma$-subgaussian coordinates and $A \in \mathbb{R}^{n \times n}$ be symmetric. Let $\|\cdot\|_F$ denote the Frobenius norm, and $\|\cdot\|$ denote the operator norm, then for all $s \geq 0$:*

$$\mathbb{P}\left(|\boldsymbol{X}^\top A \boldsymbol{X} - \mathbb{E}[\boldsymbol{X}^\top A \boldsymbol{X}]| \geq s\right) \leq 2\exp\left(-\frac{s^2/2}{40\sigma^4\|A\|_F^2 + 8s\sigma^2\|A\|}\right). \tag{10}$$

The main proof idea is to separate $A$ as a diagonal matrix $D$ and a diagonal-free matrix $F$, derive the upper bound of $\psi(t)$ of $\boldsymbol{X}^\top D \boldsymbol{X}$ and $\boldsymbol{X}^\top F \boldsymbol{X}$ separately, and use Cauchy-Schwartz inequality as:

$$\psi_{Y+Z}(t) \leq \frac{1}{2}\psi_Y(2t) + \frac{1}{2}\psi_Z(2t). \tag{11}$$

Theorem B.6 provides a general concentration result for any symmetric matrix $A$. In the special case of the sample variance, we can express it as $\hat{\sigma}_n^2 = \boldsymbol{X}^\top A \boldsymbol{X}$ with $A = \frac{1}{n-1}(I_n - \frac{1}{n}\mathbf{1}\mathbf{1}^\top)$. Leveraging this specific structure, we can apply Hölder's inequality in equation (11) to derive a more refined concentration bound of the sample variance estimator, which is a tighter version of Lemma 2.4:

**Corollary B.7.** *(A tighter version of Lemma 2.4) Suppose $X$ is $\sigma$-subgaussian, then we have:*

$$\mathbb{P}\left(\hat{\sigma}_n^2 - \sigma_X^2 \geq 4\sigma^2 f(n)\sqrt{\frac{2\log(1/\delta)}{n-1}} + \frac{6\sigma^2\log(1/\delta)}{n}\right) \leq \delta, \quad \mathbb{P}\left(\sigma_X^2 - \hat{\sigma}_n^2 \geq 4\sigma^2 f(n)\sqrt{\frac{2\log(1/\delta)}{n-1}} + \frac{13\sigma^2\log(1/\delta)}{3n}\right) \leq \delta.$$

*where $f(n) = (1 + \sqrt{n-1})/\sqrt{n}$. Specifically, if $X$ is strictly-subgaussian, then $f(n) = (1 + \sqrt{(n-1)/8})/\sqrt{n}$, and $\sigma$ can be replaced with $\sigma_X$.*

The detailed proof is in Appendix E.2. For Lemma 2.4, a slightly weaker but still informative version is:

$$\mathbb{P}\big(|\hat{\sigma}_n^2 - \sigma_X^2| \ge s\big) \le 2\exp\left(-\frac{s^2/2}{32\sigma^4/(n-1) + 6s\sigma^2/n}\right), \text{ for } s > 0.$$

Using the fact that $\|A\|_F^2 = \|A\|_2 = 1/(n-1)$ for the sample variance, this bound is tighter than the general bound (10). Additionally, using the inequality $|\hat{\sigma}_n - \sigma_X| \le \frac{|\hat{\sigma}_n^2 - \sigma_X^2|}{\sigma_X}$, with probability at least $1 - 2\delta$, we have:

$$|\hat{\sigma}_n - \sigma_X| \le \frac{4\sigma^2(1 + \sqrt{n-1})}{\sigma_X\sqrt{n}}\sqrt{\frac{2\log(1/\delta)}{n-1}} + \frac{6\sigma^2\log(1/\delta)}{n}.$$

Compared with the bound (9), this result removes one factor of $\sqrt{\log(1/\delta)}$. In particular, when setting $\delta = T^{-\alpha_0}$ with $\alpha_0 \ge 1$, as is commonly done in bandit problems, the improvement amounts to eliminating one $\sqrt{\log T}$ term.

## C. More Detailed Theoretical Analyses

In this section, we expand on the theoretical analyses of our algorithms from Sections 3 and 4, which were only briefly outlined in the main text due to space constraints. In addition, we specify the setting corresponding to a strictly subgaussian distribution. We use $\varepsilon_n^-(\delta)$ and $\varepsilon_n^+(\delta)$ represent the confidence bound shown in Corollary B.7. This notation could be simplified as $\varepsilon_n(\delta)$ if using Lemma 2.4.

In our framework, the algorithms select each $k$-th alternative for $\tau_k$ rounds ($\tau_k = \tau$ in the non-adaptive case), then estimates $\lambda_{k,\tau_k}$ for each arm, and subsequently allocates the remaining rounds according to these estimates. If we ensure that $\tau_k \le n_k$ for all $k = 1, \ldots, K$, we can define

$$\boldsymbol{\lambda} := \boldsymbol{n}/T = \{\lambda_{1,\tau_1}, \cdots, \lambda_{K,\tau_K}\}, \quad \boldsymbol{\lambda}^* := \boldsymbol{n}^*/T = \{\lambda_1^*, \cdots, \lambda_K^*\},$$

then the objective function can be expressed equivalently as:

$$R_p(\boldsymbol{n}) = R_p(\boldsymbol{\lambda}) = \frac{1}{T}\begin{cases} \left(\sum_{k=1}^K \left(\frac{\sigma_k^2}{\lambda_k}\right)^p\right)^{1/p}, & p < \infty, \\ \max_{k=1,\ldots,K} \frac{\sigma_k^2}{\lambda_k}, & p = \infty. \end{cases}$$

### C.1. Detailed Analyses for Section 3

Since in Algorithm 1, each alternative is chosen for same times, and the concentration bound at time $n$ is same for each alternative, then we can use $\varepsilon_{k,n}^-(\delta) = \varepsilon_n^-(\delta)$ and $\varepsilon_{k,n}^+(\delta) = \varepsilon_n^+(\delta)$ to denote the concentration bound for $k$-th alternative for simplicity. Since Algorithm 1 ensures that $\tau \le \min_{k=1,\ldots,K} n_k^*$, then we have $\lambda_{k,\tau_k} = \lambda_{k,\tau}$. To give a high-probability theoretical bound, as Algorithm 1 is non-adaptive, let $\delta = T^{-1}$ for infinite case and $\delta = T^{-3/2}$ for finite case, and define the event

$$\xi_\tau(\delta) = \{-\varepsilon_\tau^-(\delta) \le \hat{\sigma}_{k,\tau}^2 - \sigma_k^2 \le \varepsilon_\tau^+(\delta) \text{ for all } k\},$$

then we have $\mathbb{P}(\xi_\tau^c(\delta)) \le 2K\delta$. We begin by considering the case $p = \infty$, where the regret difference simplifies to:

$$R_\infty(\boldsymbol{\lambda}) - R_\infty(\boldsymbol{\lambda}^*) = \frac{1}{T}\left(\max_{1 \le k \le K} \frac{\sigma_k^2}{\lambda_{k,\tau}} - \Sigma_2\right).$$

Suppose $\xi_\tau(\delta)$ exits, since now the confidence radius is identical across all arms, the worst-case scenario occurs when the arm with the smallest true variance realizes its lower confidence bound, while all other arms attain their respective upper bounds. Let $\boldsymbol{\lambda}_{\pi_1} = \boldsymbol{n}_{\pi_1} \cdot T^{-1}$, then we could achieve Theorem 3.1 based on Lemma C.1.

**Lemma C.1.** *If $\xi_\tau(\delta)$ exists and $\varepsilon_\tau^-(\delta) < \sigma_{\min}^2$, then we have:*

$$R_\infty(\boldsymbol{\lambda}_{\pi_1}) - R_\infty(\boldsymbol{\lambda}^*) \le \frac{1}{T}\left[(K-1)\varepsilon_\tau^+ + \left(\frac{\Sigma_2}{\sigma_{\min}^2} - 1\right)\varepsilon_\tau^-\right] + o(T^{-3/2}).$$

When $p$ is finite, we could leveraging the smoothness property of $R_p(\boldsymbol{\lambda})$ based on Lemma C.2, which is Lemma 4 in Aznag et al. (2023). But as it is difficult to identify the worst-case configuration of the variance estimates, we give a slightly loose upper bound Theorem 3.2 based on Lemma C.3.

**Lemma C.2.** *(Lemma 4 in Aznag et al. (2023), rephrased) If $p$ is finite, then we have:*

$$R_p(\boldsymbol{\lambda}) - R_p(\boldsymbol{\lambda}^*) \leq \frac{(p+1)R_p(\boldsymbol{\lambda}^*)}{2} \sum_{k=1}^{K} \frac{(\lambda_k - \lambda_k^*)^2}{\lambda_k^*} + \frac{7(p+2)^2}{\lambda_{\min}^* T} \max_k \left(\frac{\lambda_k^*}{\lambda_k}\right)^{3p+3} \|\boldsymbol{\lambda} - \boldsymbol{\lambda}^*\|_\infty^3.$$

**Lemma C.3.** *If $\xi_\tau(\delta)$ exists and $\varepsilon_\tau^-(\delta) < \sigma_{\min}^2$, then for $p$ is finite, we have:*

$$R_p(\boldsymbol{\lambda}_{\pi_1}) - R_p(\boldsymbol{\lambda}^*) \leq \frac{p^2(\Sigma_q)^{1/p}\Sigma_{q-4}}{2(p+1)T}(\varepsilon_\tau^+)^2 + o(T^{-2}).$$

## C.2. Detailed discussion in Section 4

We begin with introducing the design of LCBs and UCBs in Section 4. Let $\delta = T^{-2}$ for infinite case and $\delta = T^{-5/2}$ for finite case, and define the event $\xi_T(\delta)$ as:

$$\xi_T(\delta) = \left\{ -\varepsilon_{k,n}^-(\delta) \leq \hat{\sigma}_{k,n}^2 - \sigma_k^2 \leq \varepsilon_{k,n}^+(\delta) \text{ for all } k \text{ and } n \right\},$$

then we have $\mathbb{P}(\xi_T^c(\delta)) \leq 2T\delta$. For each arm $k$ and time step $n$, for general subgaussian case, define the lower and upper confidence bounds for arm $k$ at time $n$ as:

$$\text{LCB}_{k,n} = \max\{\hat{\sigma}_{k,n}^2 - \varepsilon_{k,n}^+, 0\}, \quad \text{UCB}_{k,n} = \hat{\sigma}_{k,n}^2 + \varepsilon_{k,n}^-,$$

then we can get

$$\mathbb{P}\left(\text{LCB}_{k,n} \leq \sigma_k^2 \leq \text{UCB}_{k,n} \text{ for all } k \text{ and } n\right) = \mathbb{P}(\xi_\tau(\delta)) \geq 1 - 2T^{-c},$$

with $c = 1$ for $p = \infty$ and $c = 3/2$ for $p < \infty$.

For LCBs, we need to make sure $\min_k \text{LCB}_{k,n} > 0$ otherwise the second phase of Algorithm 2 would fail to work. To avoid confusion, let $\alpha_k \leq \tau_k/T$ at the end of Phase 2. Building on the analysis presented before, the key challenge in the general setting is to identify $\alpha_k$ such that $\alpha_k T \leq \lambda_k^* T$. Here we consider the general subgaussian case, the following lemma characterizes the corresponding relationship between $\alpha_k$ and $\lambda_k^*$ when $T$ is large:

**Lemma C.4.** *If $\xi_T(\delta)$ holds, then the exploration length $\tau_k \geq \alpha_k T$, with $\alpha_k = \lambda_k^*(1 - \Theta(\sqrt{T^{-1}\log T}))$.*

Then based on Lemma C.4, and follow the proof procedure of Theorem 3.1 and 3.2, we could finally achieve Theorem 4.1. In Theorem 4.1 when $p = \infty$, careful readers may notice additional factors such as $\Sigma_{-1}$ and $\sigma_{\min}^{-1}$ in our bound compared with the result in Aznag et al. (2023). These terms stem from our exploration-free design: the estimate of $\lambda_{k,\tau_k}$ is computed only after each arm has been pulled $\tau_k$ times, and $\tau_k$ is close to $\lambda_{\min}^* T$ by Lemma C.4. In the worst-case regret, we therefore include $\varepsilon_{\tau_k}$, whose leading term scales as $\sqrt{\log T/\tau_k}$, which in turn yields the factors $\Sigma_{-1}$ and $\sigma_{\min}^{-1}$. However, we need to stress that our Algorithm 2 removes one factor of $\log T$ relative to Aznag et al. (2023) beyond the refined concentration inequality, resulting in a strictly tighter asymptotic rate.

If prior information of $\underline{\sigma}^2$ is available, it still can be effectively incorporated into Algorithm 2. Rather than playing each arm twice, one may directly allocate $\tau$ rounds as prescribed by Equation (5) to each arm in the first phase, since this value serves as a deterministic lower bound for sufficient exploration.

## C.3. Strictly Subgaussian Case Analysis

For the strctly-subgaussian case, based on Lemma 2.4 we know that the concentration bound is proportional to the real variance for each arm $k$, then we can use $\varepsilon_{k,n}^-(\delta) = \sigma_k^2 \cdot s_n^-(\delta)$ and $\varepsilon_{k,n}^+(\delta) = \sigma_k^2 \cdot s_n^+(\delta)$ to represent the concentration bound of arm $k$ for simplicity, which means for $\xi_\tau(\delta)$ in Algorithm 1, we have:

$$\sigma_k^2 \cdot (1 - s_\tau^-(\delta)) \leq \hat{\sigma}_{k,\tau}^2 \leq \sigma_k^2 \cdot (1 + s_\tau^+(\delta))$$

Then we can get the following result based on Theorem 3.1 and Theorem 3.2:

**Theorem C.5.** *In the strictly-subgaussian setting of Algorithm 1, for* $p = \infty$, *let* $\delta = T^{-1}$, *we have:*

$$\mathbb{E}\left[R_p(\boldsymbol{n}_{\pi_1}) - R_p(\boldsymbol{n}^*)\right] \leq 4\lambda^{-1/2}(\Sigma_2 - \sigma_{\min}^2)T^{-3/2}\sqrt{\log T} + o(T^{-3/2}).$$

*For* $p$ *is finite, let* $\delta = T^{-3/2}$, *we have:*

$$\mathbb{E}\left[R_p(\boldsymbol{n}_{\pi_1}) - R_p(\boldsymbol{n}^*)\right] \leq \frac{3p^2(\Sigma_q)^{2/q}}{\lambda(p+1)}T^{-2}\log T + o(T^{-2}).$$

*where* $\lambda$ *is defined in Section 3.1.*

For Algorithm 2 as an adaptive process, then we could define the lower and upper confidence bound for each arm $k$ at time $n$ as:

$$\text{LCB}_{k,n} = \frac{\hat{\sigma}_{k,n}^2}{1 + s_n^+}, \quad \text{UCB}_{k,n} = \frac{\hat{\sigma}_{k,n}^2}{1 - s_n^-}.$$

Then in the event $\xi_T(\delta)$, we have:

$$\sigma_k^2 \cdot \frac{1 - s_n^-}{1 + s_n^+} \leq \text{LCB}_{k,n} \leq \sigma_k^2 \leq \text{UCB}_{k,n} \leq \sigma_k^2 \cdot \frac{1 + s_n^+}{1 - s_n^-}.$$

The particular form of our LCB and UCB has a convenient property in the first phase of Algorithm 2: it suffices to ensure

$$s_n^-(\delta) < 1.$$

Because $s_n^-(\delta)$ depends only on $(n, \delta)$, once $\delta$ is fixed we can compute the exact minimal $n$ required to meet this condition, independently of $\sigma^2$ and $\sigma_{\min}^2$. This decoupling not only simplifies the analysis and implementation but can also reduce the number of initial samples needed. For the theoretical analysis, based on Lemma 2.4 and C.4, it is easy to certify that Lemma C.4 would still be satisfied in strictly-subgaussian case. Then we have the following result:

**Theorem C.6.** *In the strictly-subgaussian setting of Algorithm 2, for* $p = \infty$, *let* $\delta = T^{-2}$, *we have:*

$$\mathbb{E}\left[R_p(\boldsymbol{n}_{\pi_2}) - R_p(\boldsymbol{n}^*)\right] \leq 2\sqrt{2}\left[\frac{\sqrt{\Sigma_2}(\Sigma_2 - 2\sigma_{\min}^2)}{\sigma_{\min}} + \sqrt{\Sigma_2}\Sigma_1\right]T^{-3/2}\sqrt{\log T} + o(T^{-3/2}).$$

*For* $p$ *is finite, let* $\delta = T^{-5/2}$, *we have:*

$$\mathbb{E}\left[R_p(\boldsymbol{n}_{\pi_2}) - R_p(\boldsymbol{n}^*)\right] \leq \frac{5Kp^2(\Sigma_q)^{2/q}}{p+1}T^{-2}\log T + o(T^{-2}).$$

As established in Section 2.1, the leading-order terms in the concentration bounds are equivalent for strictly-subgaussian and Gaussian distributions. Consequently, the regret result in the strictly-subgaussian setting remains valid for the Gaussian case, with the only modification being the substitution of $s_n^+$ and $s_n^-$ with their counterparts derived from the Gaussian-specific concentration bounds in equation 4. The difference of the results between these two cases lies solely in an asymptotically negligible residual term.

## C.4. Upper Confidence Bound in Phase 3 of Algorithm 2

Since our analysis is conducted from a worst-case perspective, and the worst-case configuration can be explicitly characterized in the case of $p = \infty$, then the inequality $1 + x \leq (1 - x)^{-1}$ indicates that replacing unknown quantities with upper confidence bounds can lead to a smaller worst-case upper bound. This observation motivates the use of a UCB-type estimator in the worst-case regime. Consequently, for $p = \infty$, we adopt a UCB estimator in the third phase of Algorithm 2 and compute $\lambda_{k,\tau_k}$ as

$$\lambda_{k,\tau_k} = \frac{(\hat{\sigma}_{k,\tau_k}^2 + \varepsilon_{k,\tau_k}^-)^{q/2}}{(\hat{\sigma}_{k,\tau_k}^2 + \varepsilon_{k,\tau_k}^-)^{q/2} + \sum_{j \neq k}(\hat{\sigma}_{j,\tau_j}^2 + \varepsilon_{j,\tau_j}^-)^{q/2}},$$

then for the general subgaussian setting, we can get a tighter value of $\mathcal{F}_{\text{Alg} 2,\infty}(\boldsymbol{\sigma^2})$:

$$\mathcal{F}_{\text{Alg} 2,\infty}(\boldsymbol{\sigma^2}) = 2\sqrt{\Sigma_2}(\Sigma_{-1} - \sigma_{\min}^{-1}),$$

and for the strictly subgaussian setting, we would have:

$$\mathbb{E}\left[R_\infty(\boldsymbol{n}_{\pi_2}) - R_\infty(\boldsymbol{n}^*)\right] \leq 4\sqrt{2}\left[\sqrt{\Sigma_2}(\Sigma_1 - \sigma_{\min})\right]T^{-3/2}\sqrt{\log T} + o(T^{-3/2}).$$

### C.5. Theoretical Result Comparison

Here we compare our theoretical results in the general subgaussian setting with those of Aznag et al. (2023). We first restate their result:

For the case $p = \infty$:

$$\mathbb{E}\left[R_p(\boldsymbol{n}) - R_p(\boldsymbol{n}^*)\right] \leq (C_T\sqrt{\Sigma_2} + C_T^2)K^{3/2}T^{-3/2} + o(T^{-3/2}).$$

For the case $p$ is finite:

$$\mathbb{E}\left[R_p(\boldsymbol{n}) - R_p(\boldsymbol{n}^*)\right] \leq \frac{8K^3 p^2 (\Sigma_q)^{2/q-2}}{p+1} \sigma_{\min}^{\frac{2(p-1)}{p+1}} C_T^2 T^{-2} + o(T^{-2}),$$

where $C_T = 16\sigma \log T + \mathcal{O}(T^{-2}\sqrt{\log T})$.

To compare the results between ours and theirs, for $p = \infty$ if we use the UCB estimator as shown in Appendix C.4 since it would give a theoretical tighter worst-case upper bound due to $1 + x \leq (1-x)^{-1}$, our result is:

For the case $p = \infty$:

$$\mathbb{E}\left[R_p(\boldsymbol{n}) - R_p(\boldsymbol{n}^*)\right] \leq 16\sigma^2\sqrt{\Sigma_2}(\Sigma_{-1} - \sigma_{\min}^{-1})T^{-3/2}\sqrt{\log T} + o(T^{-3/2}).$$

For the case $p$ is finite:

$$\mathbb{E}\left[R_p(\boldsymbol{n}) - R_p(\boldsymbol{n}^*)\right] \leq \frac{40\sigma^4 p^2 (\Sigma_q)^{2/q}(\Sigma_{-4})}{p+1}T^{-2}\log T + o(T^{-2}).$$

Comparing the results, our bound contains additional terms such as $\sigma$, $\Sigma_{-1}$, and $\Sigma_{-4}$. These arise from our refined concentration inequality and worst-case analysis, the refined concentration inequality introduces an extra factor of $\sigma/\sigma_X$ compared with the previous inequality while eliminating one $\sqrt{\log T}$ term, the worst-case analysis is related to $1/\lambda_k$ while eliminating the influence of $K$, and $\log T$ (for $p = \infty$); see the end part of Appendix B and Appendix C.2. Besides, our multiplier is 16 and 40 compared to their $16^2$ and $8 * 16^2$.

## D. More Experiments and Experiment Details

In this section, we provide some experiment details and present some additional numerical experiments.

### D.1. Experiment Details

In our numerical experiments, the initial run length per arm in the first phase of Algorithm 2 when $\underline{\sigma}^2$ is unknown is chosen as $\min\{64\sigma^4 \log T, T/K\}$ for the general subgaussian setting and $\min\{18 \log T, T/K\}$ for the strictly subgaussian setting. When $\underline{\sigma}^2$ is known, we set $\tau$ as in (5). In all cases, we then increment the number by one until the first phase stopping condition is satisfied.

Besides, in the second phase of Algorithm 2, instead of checking the condition $n_k \geq \lambda_{k,n_k} T$ after every unit increase of $n$, we use an iterative batched thresholding scheme, which reduces the number of feasibility tests and improves running time. We need to note that per-increment checks might yield slightly finer stopping times, and thus marginally more accurate empirical allocations. Here the batched scheme achieves nearly identical outcomes in practice at a substantially lower computational cost.

Finally, since we apply the floor function at each allocation update to enforce integer sample counts, which may leave a small residual budget due to rounding. After the main allocation, we greedily top up by assigning the leftover rounds sequentially to the arms with the largest estimated variances. This final correction ensures $\sum_k n_k = T$ while preserving the scheme's asymptotic optimality, since the correction is at most $K$ times and thus negligible relative to $T$.

*Table 1.* Comparison of numerical regret at $T = 10^6$ under different values of $p$.

|  | UCB | UCB-refined | Ours with GSG | Ours with SSG |
|---|---|---|---|---|
| $p = 1$ | $5.77 \times 10^{-8}$ | $1.13 \times 10^{-8}$ | $3.62 \times 10^{-10}$ | $4.15 \times 10^{-11}$ |
| $p = 2$ | $8.58 \times 10^{-8}$ | $1.66 \times 10^{-8}$ | $4.13 \times 10^{-10}$ | $6.83 \times 10^{-11}$ |
| $p = \infty$ | $7.57 \times 10^{-7}$ | $2.52 \times 10^{-7}$ | $4.61 \times 10^{-8}$ | $2.18 \times 10^{-8}$ |

### D.2. Traditional MAB: Gaussian arms

Here we provide more numerical figures as in our first Gaussian alternative experiment.

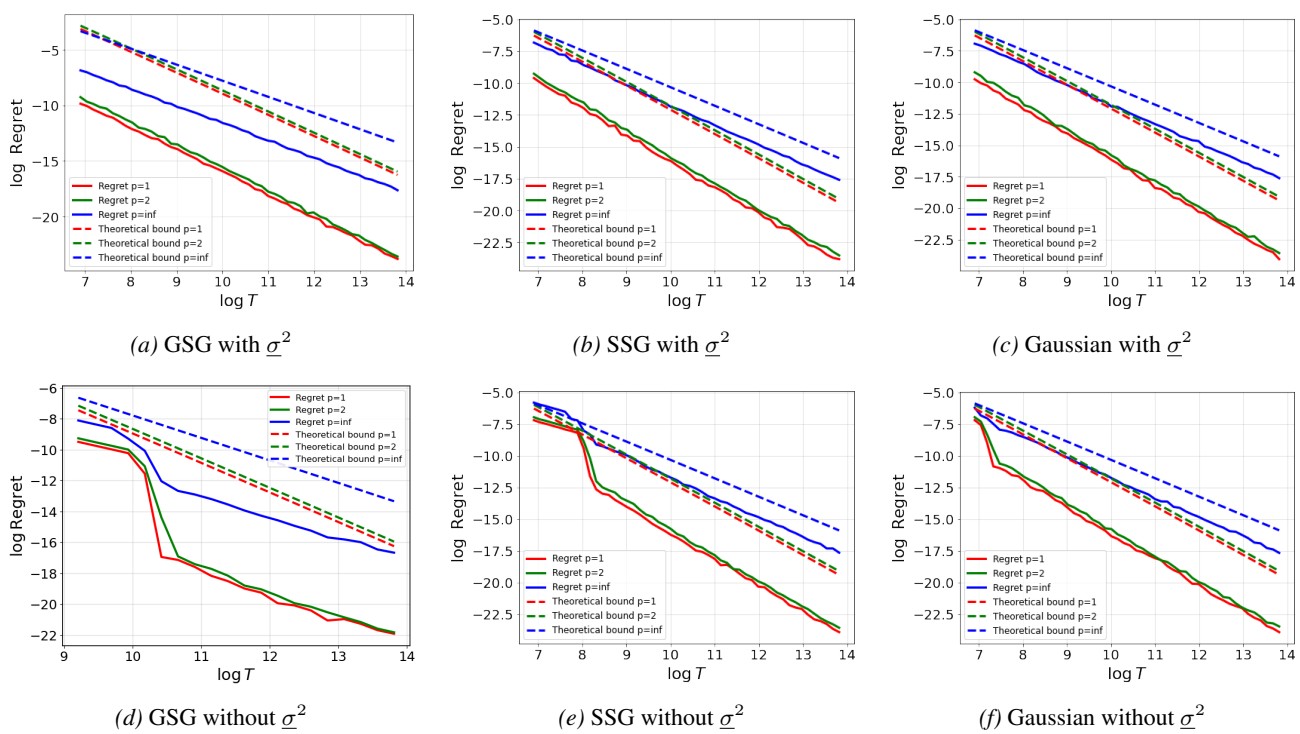

*Figure 4.* Algorithm 2: comparison of three regimes with and without $\underline{\sigma}^2$.

From Figure 4, we first observe that the performance of SSG closely resembles that of the Gaussian case, indicating that SSG effectively captures the key properties of Gaussian noise. By contrast, when $\underline{\sigma}^2$ is known, the second phase can be entered much earlier, since $\underline{\sigma}^2$ provides a lower bound for $n^*_{\min}$ and thereby circumvents the stringent exploration requirement in the first phase.

We provide a comparison between our algorithm and the UCB-based methods studied in Aznag et al. (2023). Specifically, *UCB* corresponds to the original algorithm in Aznag et al. (2023) with their original concentration inequality, while *UCB-refined* applies the same algorithm equipped with our improved concentration inequality in the general subgaussian setting. The remaining methods correspond to our proposed algorithms under different regimes.

Figure 5 illustrates the asymptotic regret behavior of these algorithms for different values of $p$ in our Gaussian alternatives experiment, where each curve reports the mean regret over 100 independent replications. Table 1 reports the mean numerical regret at a fixed horizon $T = 10^6$ over the same 100 independent replications, which provides a quantitative comparison of the absolute performance levels across different values of $p$.

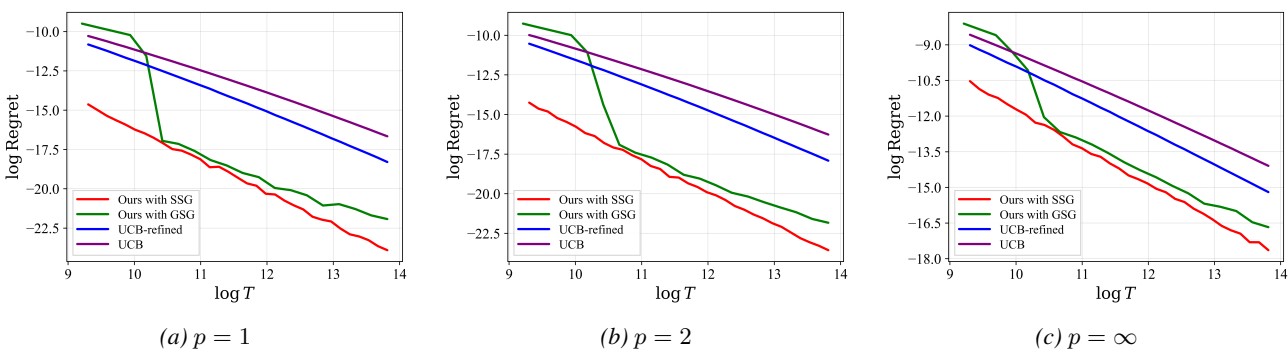

*(a)* $p = 1$          *(b)* $p = 2$          *(c)* $p = \infty$

*Figure 5.* Comparison between Algorithm 2 and UCB-based methods under different concentration inequalities.

### D.3. Another Strictly Subgaussian Example: Symmetric Beta

To further validate the benefits of the strictly-subgaussian property, we consider a non-Gaussian example based on the symmetric Beta distribution $\text{Beta}(\alpha, \alpha)$, supported on $[0, 1]$. This family is flexible: for $\alpha < 1$ it is U-shaped, while larger $\alpha$ values concentrate mass near $0.5$, with variance $(2\alpha + 1)^{-1}$, and it is known to satisfy strictly-subgaussian properties, making it a natural candidate for studying group mean estimation.

In our experiment, we simulate $K = 4$ arms with rewards $X_{k,n} = \mu_k + \text{Beta}(\alpha, \alpha)$, where $\mu_k \sim \mathcal{U}([-1, 1])$. The shape parameters $\{\alpha_k\} = \{0.2, 1.0, 2.0, 4.5\}$ are chosen to cover a wide spectrum of tail behaviors. We set $\underline{\sigma}^2 = \sigma_{\min}^2$ and $\sigma^2 = 1$ from the variance formula.

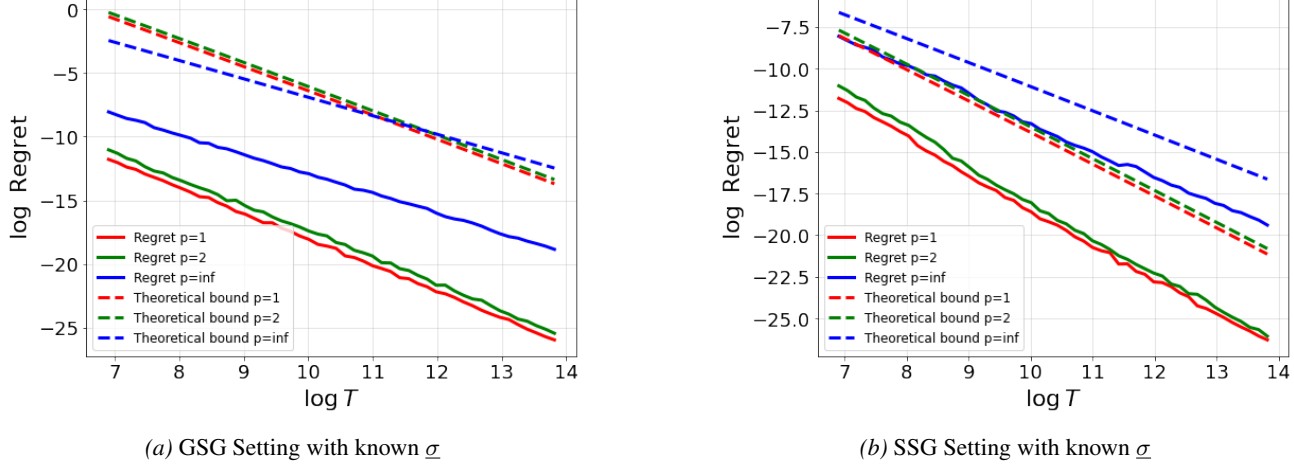

*(a)* GSG Setting with known $\underline{\sigma}$          *(b)* SSG Setting with known $\underline{\sigma}$

*Figure 6.* Algorithm 2: Symmetric Beta.

## E. Proof of Section 2

This section gives the proof of Proposition 2.1 and Lemma 2.4 in Section 2.

### E.1. Proof of Proposition 2.1

*Proof.* For the case $p < \infty$, since $f(x) = x^{1/p}$ is non-decreasing, it is equivalent to minimizing $\sum_{k=1}^{K} \sigma_k^{2p}/n_k^p$ under $\sum_{k=1}^{K} n_k = T$. By using the Lagrangian and the first-order optimality conditions, we can get that $n_k^{p+1} \propto \sigma_k^{2p}$, therefore we have:

$$n_k^* = \frac{\sigma_k^{\frac{2p}{p+1}}}{\sum_{j=1}^{K} \sigma_j^{\frac{2p}{p+1}}} T = \frac{\sigma_k^q}{\sum_{j=1}^{K} \sigma_j^q} T,$$

then plug into $R_p$, we have:

$$R_p(\boldsymbol{n}^*) = \left[ \left( \frac{\sum_j \sigma_j^q}{T} \right)^p \sum_k \sigma_k^q \right]^{1/p} = \frac{1}{T} \left( \sum_{k=1}^K \sigma_k^q \right)^{\frac{p+1}{p}} = \frac{1}{T} \left( \sum_{k=1}^K \sigma_k^q \right)^{\frac{2}{q}}.$$

For the case $p = \infty$, the objective is equivalent to minimize $t$ such that $\sigma_k^2/n_k \leq t$ for all $k$ and $\sum_{k=1}^K n_k = T$. Then we can get $n_k \geq \sigma_k^2/t$, hence $t \geq \sum_{k=1}^K \sigma_k^2/T$. Then we could get that:

$$n_k^* = \frac{\sigma_k^2}{\sum_{j=1}^K \sigma_j^2} T, \quad R_\infty(\boldsymbol{n}^*) = \frac{1}{T} \sum_{k=1}^K \sigma_k^2.$$

$\square$

## E.2. Proof of Lemma 2.4

*Proof.* The proof is based on Section 2 of Boucheron et al. (2013), Section 6 of Vershynin (2018) and the theoretical framework of Epperly (2022). For the sample variance, let $D = \frac{1}{n} I_n$ and $F = -\frac{1}{n(n-1)} \mathbf{1}\mathbf{1}^\top + \frac{1}{n(n-1)} I_n$, then we have $\hat{\sigma}_n^2 = \boldsymbol{X}^\top A\boldsymbol{X} = \boldsymbol{X}^\top D\boldsymbol{X} + \boldsymbol{X}^\top F\boldsymbol{X}$ and $\mathbb{E}[\boldsymbol{X}^\top A\boldsymbol{X}] = \mathbb{E}[\boldsymbol{X}^\top D\boldsymbol{X}]$.

We first show the right-tail bound when $X$ belongs to $\sigma$-subgaussian. For the diagonal matrix $D$, since each entry $d_{i,i} = n^{-1}$, then for $t > 0$, based on Lemma B.2, we have:

$$\psi_{\boldsymbol{X}^\top D\boldsymbol{X} - \mathbb{E}[\boldsymbol{X}^\top A\boldsymbol{X}]}(t) = \frac{8\sigma^4 t^2 n^{-1}}{1 - 2\sigma^2 t/n}$$

For the diagonal-free matrix $F$, we apply the decoupling bound together with the standard Gaussian quadratic form representation $\tilde{g}^\top F g$ (see Theorem 6.1.1 and Lemma 6.2.3 in Vershynin (2018), Epperly (2022)) to obtain:

$$\psi_{\boldsymbol{X}^\top F\boldsymbol{X}}(t) \leq \psi_{\tilde{g}^\top Fg}(4\sigma^2 t) \leq \frac{8\sigma^4 \|F\|_F^2 t^2}{1 - 4\sigma^2 \|F\| t} = \frac{8\sigma^4 t^2/(n^2 - n)}{1 - 4\sigma^2 t/n},$$

where the second inequality is by Hermitian dilation: $\psi_{\tilde{g}^\top Fg}(t) \leq \frac{\|F\|_F^2 t^2}{2(1 - \|F\|t)}$, the last equality is from $\|F\|_F^2 = 1/(n^2 - n)$ and $\|F\| = 1/n$. Then we can apply Hölder's inequality, let positive $p, q$ satisfy $1/p + 1/q = 1$:

$$\begin{aligned}
\psi_{\boldsymbol{X}^\top A\boldsymbol{X} - \mathbb{E}[\boldsymbol{X}^\top A\boldsymbol{X}]}(t) &\leq \frac{1}{p} \psi_{\boldsymbol{X}^\top D\boldsymbol{X} - \mathbb{E}[\boldsymbol{X}^\top A\boldsymbol{X}]}(pt) + \frac{1}{q} \psi_{\boldsymbol{X}^\top F\boldsymbol{X}}(qt) \\
&= \frac{8\sigma^4 pt^2 n^{-1}}{1 - 2\sigma^2 pt/n} + \frac{8\sigma^4 qt^2/(n^2 - n)}{1 - 4\sigma^2 qt/n} \\
&= 8\sigma^4 t^2 \left[ \frac{1}{n/p - 2t\sigma^2} + \frac{1}{(n-1)[(1 - 1/p)n - 4\sigma^2 t]} \right] \\
&= \frac{8\sigma^4 t^2 (1 + \sqrt{n-1})^2/(n^2 - n)}{1 - 6\sigma^2 t/n}
\end{aligned}$$

where for the last equality, let $f(x) = \frac{1}{nx - a} + \frac{C}{(n-1)[(1-x)n - b]}$, after some elementary calculation, we can get that when $n > a + b$, with $x^* = \frac{(n-b)\sqrt{(n-1)/C} + a}{n(1 + \sqrt{(n-1)/C})} < 1$, $\min f(x) = f(x^*) = \frac{(1 + \sqrt{C/(n-1)})^2}{n - (a+b)} = \frac{C(1 + \sqrt{(n-1)/C})^2/(n^2 - n)}{1 - (a+b)/n}$, then we can use $1/p = x^*$.

This implies $\boldsymbol{X}^\top A\boldsymbol{X} - \mathbb{E}[\boldsymbol{X}^\top A\boldsymbol{X}]$ belongs to $\Gamma_+(\nu, c)$ with $\nu = 16\sigma^4 (1 + \sqrt{n-1})^2/(n^2 - n)$ and $c = 6\sigma^2/n$. By Chernoff's inequality (see Section 2.4 in Boucheron et al. (2013)), if $X \in \Gamma_+(\nu, c)$ then for any $s > 0$, $\mathbb{P}(X > \sqrt{2\nu s} + cs) \leq e^{-s}$. Then set $s = \log(1/\delta)$, we can get:

$$\mathbb{P}\left( \hat{\sigma}_n^2 - \sigma_X^2 \geq \frac{4\sigma^2 (1 + \sqrt{n-1})}{\sqrt{n}} \sqrt{\frac{2\log(1/\delta)}{n-1}} + \frac{6\sigma^2 \log(1/\delta)}{n} \right) \leq \delta.$$

For the left-tail bound, for $t > 0$, since $-\boldsymbol{X}^\top A\boldsymbol{X} + \mathbb{E}[\boldsymbol{X}^\top A\boldsymbol{X}] = -\boldsymbol{X}^\top D\boldsymbol{X} + \mathbb{E}[\boldsymbol{X}^\top A\boldsymbol{X}] + \boldsymbol{X}^\top(-F)\boldsymbol{X}$, then based on Lemma B.2, $X$ belongs to $\Gamma_-(16\sigma^4, \sigma^2/3)$, we have:

$$\psi_{\boldsymbol{X}^\top(-D)\boldsymbol{X}+\mathbb{E}[\boldsymbol{X}^\top A\boldsymbol{X}]}(t) \leq \frac{8\sigma^4 t^2}{n - \sigma^2 t/3}, \quad \psi_{\boldsymbol{X}^\top(-F)\boldsymbol{X}}(t) \leq \psi_{\tilde{g}^\top(-F)g}(4\sigma^2 t) \leq \frac{8\sigma^4 t^2/(n^2 - n)}{1 - 4\sigma^2 t/n}.$$

Then we can write the cumulant generating function as:

$$\begin{aligned}
\psi_{-\boldsymbol{X}^\top A\boldsymbol{X}+\mathbb{E}[\boldsymbol{X}^\top A\boldsymbol{X}]}(t) &\leq \frac{1}{p}\psi_{-\boldsymbol{X}^\top D\boldsymbol{X}+\mathbb{E}[\boldsymbol{X}^\top A\boldsymbol{X}]}(pt) + \frac{1}{q}\psi_{\boldsymbol{X}^\top(-F)\boldsymbol{X}}(qt) \\
&= \frac{8\sigma^4 pt^2 n^{-1}}{1 - \sigma^2 pt/3n} + \frac{8\sigma^4 t^2/(n^2 - n)}{1 - 4\sigma^2 t/n} \\
&= 8\sigma^4 t^2 \left[ \frac{1}{n/p - \sigma^2 t/3} + \frac{1}{(n-1)[(1-1/p)n - 4\sigma^2 t]} \right] \\
&= \frac{8\sigma^4 t^2(1 + \sqrt{n-1})^2/(n^2 - n)}{1 - 13\sigma^2 t/3n}
\end{aligned}$$

Then $\boldsymbol{X}^\top A\boldsymbol{X} - \mathbb{E}[\boldsymbol{X}^\top A\boldsymbol{X}]$ belongs to $\Gamma_-(\nu, c)$ with $\nu = 16\sigma^4(1 + \sqrt{n-1})^2/(n^2 - n)$, $c = 13\sigma^2/3n$. Then we can get the result by using Chernoff's inequality.

For the strcitly-subgaussian case, the proof is similar, now we have $X^2 - \mathbb{E}[X^2]$ belongs to $\Gamma_+(2\sigma_X^4, 2\sigma_X^2)$ and $\Gamma_-(2\sigma_X^4, \sigma_X^2/3)$ as proved in Lemma B.2. Following the proof above, we can get that $\boldsymbol{X}^\top A\boldsymbol{X} - \mathbb{E}[\boldsymbol{X}^\top A\boldsymbol{X}]$ belongs to $\Gamma_+(\nu, c^+)$ and $\Gamma_-(\nu, c^-)$ with $\nu = 16\sigma_X^4(1 + \sqrt{(n-1)/8})^2/(n^2 - n)$, $c^+ = 6\sigma_X^2/n$ and $c^- = 13\sigma_X^2/3n$. $\square$

# F. Proof of Section 3

This section gives the proof of Theorem 3.1 and Theorem 3.2 in Section 3.

## F.1. Proof of Theorem 3.1

*Proof.* Since $\delta = T^{-1}$, we have $\mathbb{P}(\xi_\tau^c(\delta)) = 2KT^{-1}$, then:

$$\begin{aligned}
\mathbb{E}[R_\infty(\boldsymbol{n}_{\pi_1}) - R_\infty(\boldsymbol{n}^*)] &= \mathbb{E}[R_\infty(\boldsymbol{\lambda}) - R_\infty(\boldsymbol{\lambda}^*)] \\
&= \mathbb{P}(\xi_\tau(\delta))\mathbb{E}[R_\infty(\boldsymbol{\lambda}) - R_\infty(\boldsymbol{\lambda}^*)|\xi_\tau(\delta)] + \mathbb{P}(\xi_\tau^c(\delta))\mathbb{E}[R_\infty(\boldsymbol{\lambda}) - R_\infty(\boldsymbol{\lambda}^*)|\xi_\tau^c(\delta)] \\
&\leq \mathbb{E}[R_\infty(\boldsymbol{\lambda}) - R_\infty(\boldsymbol{\lambda}^*)|\xi_\tau(\delta)] + 2KT^{-1}\mathbb{E}[R_\infty(\boldsymbol{\lambda}) - R_\infty(\boldsymbol{\lambda}^*)|\xi_\tau^c(\delta)] \\
&\leq \frac{1}{T}\left[(K-1)\varepsilon_\tau^+ + (\frac{\Sigma_2}{\sigma_{\min}^2} - 1)\varepsilon_\tau^-\right] + o(T^{-3/2}) \\
&\leq \frac{4\sqrt{2}\sigma^2(K + \Sigma_2/\sigma_{\min}^2 - 2)}{T}\sqrt{\frac{\log T}{\tau}} + o(T^{-3/2}) \\
&\leq 4\sqrt{2}\sigma^2\lambda^{-1/2}(K + \Sigma_2/\underline{\sigma}^2 - 2)T^{-3/2}\sqrt{\log T} + o(T^{-3/2})
\end{aligned}$$

where the second inequality is based on Lemma C.1. $\square$

## F.2. Proof of Theorem 3.2

*Proof.* Since $\delta = T^{-3/2}$, we have $\mathbb{P}(\xi_\tau^c(\delta)) = 2KT^{-3/2}$, then:

$$\begin{aligned}
\mathbb{E}[R_p(\boldsymbol{n}_{\pi_1}) - R_p(\boldsymbol{n}^*)] &= \mathbb{E}[R_p(\boldsymbol{\lambda}) - R_p(\boldsymbol{\lambda}^*)] \\
&= \mathbb{P}(\xi_\tau(\delta))\mathbb{E}[R_p(\boldsymbol{\lambda}) - R_p(\boldsymbol{\lambda}^*)|\xi_\tau(\delta)] + \mathbb{P}(\xi_\tau^c(\delta))\mathbb{E}[R_p(\boldsymbol{\lambda}) - R_p(\boldsymbol{\lambda}^*)|\xi_\tau^c(\delta)] \\
&\leq \mathbb{E}[R_p(\boldsymbol{\lambda}) - R_p(\boldsymbol{\lambda}^*)|\xi_\tau(\delta)] + 2KT^{-3/2}\mathbb{E}[R_p(\boldsymbol{\lambda}) - R_p(\boldsymbol{\lambda}^*)|\xi_\tau^c(\delta)] \\
&\leq \frac{p^2(\Sigma_q)^{1/p}\Sigma_{q-4}}{2(p+1)T}(\varepsilon_\tau^+)^2 + o(T^{-2}) \\
&= \frac{24\sigma^4 p^2(\Sigma_q)^{1/p}\Sigma_{q-4}}{\lambda(p+1)}T^{-2}\log T + o(T^{-2})
\end{aligned}$$

where the second inequality is based on Lemma C.3. □

# G. Proof of Section 4

This section gives the proof of Theorem 4.1 in Section 4.

## G.1. Proof of Theorem 4.1

*Proof.* For the case $p = \infty$, assuming $\sigma_{\min}^2 = \sigma_K^2$, then we have:

$$\max_k \frac{\sigma_k^2}{\lambda_{k,\pi_2}} - \Sigma_2 \leq \max_k \sum_{j \neq k} (\sigma_j^2 + \varepsilon_{\tau_j}^+)\left(1 + \sum_{n=1}^{\infty} \left(\frac{\varepsilon_{\tau_k}^-}{\sigma_k^2}\right)^n\right) - (\Sigma_2 - \sigma_k^2)$$

$$= \sum_{j \neq K} \varepsilon_{\tau_j}^+ + \left(\frac{\Sigma_2}{\sigma_{\min}^2} - 1\right)\varepsilon_{\tau_K}^- + \mathcal{O}(T^{-1}\log T)$$

$$\leq 8\sigma^2 \sum_{j \neq K} \sqrt{\frac{\log T}{\alpha_j T}} + 8\left(\frac{\Sigma_2}{\sigma_{\min}^2} - 1\right)\sigma^2 \sqrt{\frac{\log T}{\alpha_K T}} + \mathcal{O}(T^{-1}\log T)$$

$$= 8\sigma^2 \left[\sum_{j \neq K} \alpha_j^{-1/2} + \left(\frac{\Sigma_2}{\sigma_{\min}^2} - 1\right)\alpha_K^{-1/2}\right]\sqrt{\frac{\log T}{T}} + \mathcal{O}(T^{-1}\log T)$$

$$\leq 8\sigma^2 \sqrt{T^{-1}\log T}\left[\sqrt{\Sigma_2}\Sigma_{-1} + (\Sigma_2\sigma_{\min}^{-2} - 2)\sqrt{\Sigma_2}\sigma_{\min}^{-1}\right] + \mathcal{O}(T^{-1}\log T)$$

where the second inequality is from $\alpha_k T \leq \tau_k$, the last inequality is based on Lemma C.4 and $[1 - \Theta(\sqrt{T^{-1}\log T})]^{-1/2} = 1 + \Theta(\sqrt{T^{-1}\log T})$. Then we can get that:

$$\mathbb{E}\left[R_\infty(\boldsymbol{n}_{\pi_2}) - R_\infty(\boldsymbol{n}^*)\right] = \mathbb{E}\left[R_\infty(\boldsymbol{\lambda}) - R_\infty(\boldsymbol{\lambda}^*)\right]$$

$$= \mathbb{P}(\xi_T(\delta))\mathbb{E}\left[R_\infty(\boldsymbol{\lambda}) - R_\infty(\boldsymbol{\lambda}^*)|\xi_T(\delta)\right] + \mathbb{P}(\xi^c(\delta))\mathbb{E}\left[R_\infty(\boldsymbol{\lambda}) - R_\infty(\boldsymbol{\lambda}^*)|\xi_T^c(\delta)\right]$$

$$\leq \mathbb{E}\left[R_\infty(\boldsymbol{\lambda}) - R_\infty(\boldsymbol{\lambda}^*)|\xi_T(\delta)\right] + 2T^{-1}\mathbb{E}\left[R_\infty(\boldsymbol{\lambda}) - R_\infty(\boldsymbol{\lambda}^*)|\xi_T^c(\delta)\right]$$

$$\leq 8\sigma^2 \sqrt{T^{-3}\log T}(\sqrt{\Sigma_2}\Sigma_{-1} + (\Sigma_2\sigma_{\min}^{-2} - 2)\sqrt{\Sigma_2}\sigma_{\min}^{-1}) + o(T^{-3/2})$$

$$= 8\sigma^2 \mathcal{F}_{\text{Alg }2,\infty}(\boldsymbol{\sigma^2}) \cdot T^{-3/2}\sqrt{\log T} + o(T^{-3/2})$$

When $p$ is finite, then based on the proof of Lemma C.3, when $\xi_T(\delta)$ exists, we have:

$$\sum_{k=1}^{K} \frac{(\lambda_{k,\pi_2} - \lambda_k^*)^2}{\lambda_k^*} \leq \frac{\Sigma_q}{(\sum_{k=1}^{K}(\sigma_k^2 - \varepsilon_{\tau_k}^-)^{q/2})^2} \sum_{k=1}^{K} \frac{[(\sigma_k^2 + \varepsilon_{\tau_k}^+)^{q/2} - \sigma_k^q]^2}{\sigma_k^q}$$

$$\leq \frac{q^2 \sum_{k=1}^{K} \sigma_k^{q-4}(\varepsilon_{\tau_k}^+)^2}{4\Sigma_q} + \mathcal{O}(T^{-3/2}\log^{3/2}T)$$

$$\leq \frac{20\sigma^4 q^2 \log T}{\Sigma_q T} \sum_{k=1}^{K} \frac{\sigma_k^{q-4}}{\alpha_k} + \mathcal{O}(T^{-3/2}\log^{3/2}T)$$

$$\leq \frac{20\sigma^4 q^2 \log T}{\Sigma_q T} \sum_{k=1}^{K} \frac{\sigma_k^{q-4}}{\lambda_k^*} + \mathcal{O}(T^{-3/2}\log^{3/2}T)$$

$$= 20\sigma^4 q^2 (\Sigma_{-4})T^{-1}\log T + \mathcal{O}(T^{-3/2}\log^{3/2}T)$$

Then we have:

$$
\begin{aligned}
\mathbb{E}\left[R_\infty(\boldsymbol{n}_{\pi_2}) - R_\infty(\boldsymbol{n}^*)\right] &= \mathbb{E}\left[R_\infty(\boldsymbol{\lambda}) - R_\infty(\boldsymbol{\lambda}^*)\right] \\
&= \mathbb{P}(\xi_T(\delta))\mathbb{E}\left[R_\infty(\boldsymbol{\lambda}) - R_\infty(\boldsymbol{\lambda}^*)|\xi_T(\delta)\right] + \mathbb{P}(\xi_T^c(\delta))\mathbb{E}\left[R_\infty(\boldsymbol{\lambda}) - R_\infty(\boldsymbol{\lambda}^*)|\xi_T^c(\delta)\right] \\
&\leq \mathbb{E}\left[R_\infty(\boldsymbol{\lambda}) - R_\infty(\boldsymbol{\lambda}^*)|\xi_T(\delta)\right] + 2T^{-3/2}\mathbb{E}\left[R_\infty(\boldsymbol{\lambda}) - R_\infty(\boldsymbol{\lambda}^*)|\xi_T^c(\delta)\right] \\
&\leq \frac{40\sigma^4 p^2 (\Sigma_q)^{2/q}(\Sigma_{-4})}{p+1} T^{-2}\log T + o(T^{-2}) \\
&= 40\sigma^4 \mathcal{F}_{\mathrm{Alg}\,2,p}(\boldsymbol{\sigma}^2) \cdot T^{-2}\log T + o(T^{-2})
\end{aligned}
$$

$\square$

# H. Proof of Section 5

This section gives the proof of Lemma 5.2, Lemma 5.4 and Theorem 5.5 in Section 5.

## H.1. Proof of Lemma 5.2

*Proof.*

$$
\begin{aligned}
\hat{\beta}_{k,t} - \beta_k &= \left(\gamma I_d + \sum_{s=1}^t c_{k,s}c_{k,s}^\top\right)^{-1}\sum_{s=1}^t c_{k,s}X_{k,s} - \beta_k \\
&= \left(\gamma I_d + \sum_{s=1}^t c_{k,s}c_{k,s}^\top\right)^{-1}\sum_{s=1}^t c_{k,s}(\beta_k^\top c_{k,s} + \eta_{k,s}) - \beta_k \\
&= V_{k,t}^{-1}\sum_{s=1}^t \eta_{k,s}c_{k,s} - \gamma V_{k,t}^{-1}\beta_k.
\end{aligned}
$$

Let $\boldsymbol{\eta_{k,t}} = (\eta_{k,1}, \cdots, \eta_{k,t})^\top \in \mathbb{R}^t$, and $\boldsymbol{C_{k,t}} = [c_{k,1}, \cdots, c_{k,t}] \in \mathbb{R}^{d\times t}$, then conditioned on $\boldsymbol{C_{k,t}}$, we have:

$$
\begin{aligned}
\mathbb{E}\left[\|\hat{\beta}_{k,t} - \beta_k\|^2 \mid \boldsymbol{C_{k,t}}\right] &= \mathbb{E}\left[\left\|V_{k,t}^{-1}\sum_{s=1}^t \eta_{k,s}c_{k,s} - \gamma V_{k,t}^{-1}\beta_k\right\|^2 \middle| \boldsymbol{C_{k,t}}\right] \\
&= \mathbb{E}\left[\gamma^2\beta_k^\top V_{k,t}^{-2}\beta_k + \boldsymbol{\eta_{k,t}}^\top \boldsymbol{C_{k,t}^\top} V_{k,t}^{-2}\boldsymbol{C_{k,t}}\boldsymbol{\eta_{k,t}} - 2\boldsymbol{\eta_{k,t}}^\top \boldsymbol{C_{k,t}^\top} V_{k,t}^{-2}\gamma\beta_k \middle| \boldsymbol{C_{k,t}}\right] \\
&= \mathbb{E}\left[\gamma^2\beta_k^\top V_{k,t}^{-2}\beta_k + \boldsymbol{\eta_{k,t}}^\top \boldsymbol{C_{k,t}^\top} V_{k,t}^{-2}\boldsymbol{C_{k,t}}\boldsymbol{\eta_{k,t}} \middle| \boldsymbol{C_{k,t}}\right] \\
&= \gamma^2\beta_k^\top V_{k,t}^{-2}\beta_k + \sigma_k^2 \operatorname{Tr}(V_{k,t}^{-2}\boldsymbol{C_{k,t}}\boldsymbol{C_{k,t}^\top}) \\
&= \gamma^2\beta_k^\top V_{k,t}^{-2}\beta_k + \sigma_k^2 \operatorname{Tr}(V_{k,t}^{-2}(V_{k,t} - \gamma I_d)) \\
&= \gamma^2\beta_k^\top V_{k,t}^{-2}\beta_k + \sigma_k^2\left(\operatorname{Tr}(V_{k,t}^{-1}) - \gamma\operatorname{Tr}(V_{k,t}^{-2})\right) \\
&= \sigma_k^2 \operatorname{Tr}(V_{k,t}^{-1}) + \gamma^2\beta_k^\top V_{k,t}^{-2}\beta_k - \gamma\sigma_k^2 \operatorname{Tr}(V_{k,t}^{-2}).
\end{aligned}
$$

$\square$

## H.2. Proof of Lemma 5.4

*Proof.* First let $V_{k,n}(0) = \sum_{s=1}^n c_{k,s}c_{k,s}^\top$ when $\gamma = 0$. As we set $\gamma = \lambda_{\min}^{\mathcal{C}}/n$, we note that for $V_{k,n}$ we can decide whether to add $\gamma I_d$ or not based on $\sum_{s=1}^n c_{k,s}c_{k,s}^\top$ as:

$$
V_{k,n} = \left[\gamma - \lambda_{\min}(V_{k,n}(0))\right]\mathbb{I}_{\{\lambda_{\min}(V_{k,n}(0))<\gamma\}}I_d + V_{k,n}(0).
$$

This means that only when $\lambda_{\min}(V_{k,n}(0))$ is very small, we need to add this term. Let $\zeta_{k,n} = \{\lambda_{\min}(V_{k,n}(0)) \geq \gamma\}$, then based on Lemma 5.2, we have:

$$
\mathbb{E}\left[\|\hat{\beta}_{k,n} - \beta_k\|^2\right] \leq \mathbb{P}\left(\zeta_{k,n}^c\right) \cdot \left[\gamma^2\|\beta_k\|^2\|V_{k,n}^{-2}\| + \sigma_k^2 \cdot \mathbb{E}\left[\mathrm{Tr}(V_{k,n}^{-1}) \,\Big|\, \zeta_{k,n}^c\right]\right]
$$
$$
+ \mathbb{P}\left(\zeta_{k,n}\right) \cdot \sigma_k^2 \cdot \mathbb{E}\left[\mathrm{Tr}\left((V_{k,n}(0))^{-1}\right) \,\Big|\, \zeta_{k,n}\right]
$$
$$
\leq \mathbb{P}\left(\zeta_{k,n}^c\right) \cdot \left[\|\beta_k\|^2 + \sigma_k^2 \cdot \frac{d}{\gamma}\right] + \sigma_k^2 \cdot \mathbb{E}\left[\mathrm{Tr}\left((V_{k,n}(0))^{-1}\right) \,\Big|\, \zeta_{k,n}\right]. \tag{12}
$$

where the second inequality is because $\beta^\top A\beta \leq \|A\|\|\beta\|^2$ and $\|A^{-2}\| = \|A^{-1}\|^2 \leq \lambda_{\min}^{-2}(A)$ for any positive definite matrix $A$.

Then we need to bound $\lambda_{\min}(V_{k,n}(0))$ by using the matrix concentration inequality as Theorem 5.3. Let $X_s = c_sc_s^\top - \Sigma$, then $V_t(0) = t\Sigma + \sum_{s=1}^t X_s$. The matrices $\{X_s\}_{s=1}^t$ are independent, centered, self–adjoint, and satisfy $\|X_s\| \leq \|c_s\|^2 + \|\Sigma\| \leq R^2 + R^2 = 2R^2$. And we have:

$$
\left\|\mathbb{E}[X_s^2]\right\| = \left\|\mathbb{E}[(c_sc_s^\top)^2] - \Sigma^2\right\|
$$
$$
\leq \left\|\mathbb{E}[(c_sc_s^\top)^2]\right\| + \left\|\Sigma^2\right\|
$$
$$
\leq R^2\|\Sigma\| + \|\Sigma^2\|
$$
$$
\leq 2R^2\|\Sigma\|
$$

where the first equality is due to $\mathbb{E}[c_sc_s^\top\Sigma] = \Sigma^2$, the second inequality is because $\mathbb{E}[(c_sc_s^\top)^2] \preceq R^2\mathbb{E}[c_sc_s^\top]$ based on Assumption 5.1, the third inequality is from $\|\Sigma^2\| = \|\Sigma\|^2 \leq R^2\|\Sigma\|$.

Then we have $v = \|\sum_{s=1}^t \mathbb{E}[X_s^2]\| \leq 2tR^2\|\Sigma\|$. Let $S_t = \sum_{s=1}^t X_s$, and assume $m \geq 2$, by Theorem 5.3 we have:

$$
\mathbb{P}(\{\|S_t\| \geq \tfrac{m-1}{m}t\lambda_{\min}(\Sigma)\}) \leq 2d \cdot \exp(-\frac{\frac{(m-1)^2}{2m^2}t^2\lambda_{\min}^2(\Sigma)}{2tR^2\|\Sigma\| + \frac{2(m-1)}{3m}R^2t\lambda_{\min}(\Sigma)})
$$
$$
\leq 2d \cdot \exp(-\frac{3(m-1)^2t\lambda_{\min}^2(\Sigma)}{4m(4m-1)R^2\|\Sigma\|})
$$
$$
\leq 2d \cdot \exp(-t\kappa(m))
$$

Then based on the fact that $\lambda_{\min}(V_{k,n}(0)) = \lambda_{\min}(n\Sigma + S_n) \geq n\lambda_{\min}(\Sigma) - \|S_n\|$, we have:

$$
\mathbb{P}(\zeta_n^c) \leq \mathbb{P}(\{\|S_n\| \geq (1 - n^{-2})n\lambda_{\min}(\Sigma)\})
$$
$$
\leq 2d\exp(-n \cdot \kappa(n^2))
$$
$$
\leq 2d\exp(-\frac{3n\lambda_{\min}^2(\Sigma)}{56R^2\|\Sigma\|})
$$

where the last inequality is based on $n \geq 2$. When $\zeta_n$ exists, let $u_0 = 2/(n\lambda_{\min}(\Sigma))$, we can get:

$$
\mathbb{E}\left[\mathrm{Tr}\left(V_n(0)^{-1}\right) \,\Big|\, \zeta_n\right] \leq \mathbb{E}\left[\frac{d}{\lambda_{\min}(V_t(0))} \,\Big|\, \zeta_t\right]
$$
$$
= d\int_0^{u_0} \mathbb{P}(\lambda_{\min}(V_n(0)) \leq \frac{1}{u})du + d\int_{u_0}^{\frac{n}{\lambda_{\min}(\Sigma)}} \mathbb{P}(\lambda_{\min}(V_n(0)) \leq \frac{1}{u})du
$$
$$
\leq \frac{2d}{n\lambda_{\min}(\Sigma)} + \frac{d}{n\lambda_{\min}(\Sigma)}\int_2^{n^2} \mathbb{P}(\lambda_{\min}(V_n(0)) \leq \frac{n}{m}\lambda_{\min}(\Sigma))dm
$$
$$
\leq \frac{2d}{n\lambda_{\min}(\Sigma)} + \frac{d}{n\lambda_{\min}(\Sigma)}\int_2^{n^2} \mathbb{P}(\|S_n\| \geq \tfrac{m-1}{m}n\lambda_{\min}(\Sigma))dm
$$
$$
\leq \frac{2d}{n\lambda_{\min}(\Sigma)} + \frac{2d^2}{n\lambda_{\min}(\Sigma)}\int_2^{n^2} \exp(-\kappa(m)n)dm
$$
$$
\leq \frac{2d}{n\lambda_{\min}(\Sigma)} + \frac{2d^2(n^2 - 2)}{n\lambda_{\min}(\Sigma)}\exp(-\frac{3n\lambda_{\min}^2(\Sigma)}{56R^2\|\Sigma\|}).
$$

Then combine them into equation (12), and let $\lambda_{\min}^{\mathcal{C}} = \lambda_{\min}(\Sigma)$, we have:

$$
\begin{aligned}
\mathbb{E}\left[\|\hat{\beta}_{k,n} - \beta_k\|^2\right] &\leq 2d \cdot \exp(-\frac{3n(\lambda_{\min}^{\mathcal{C}})^2}{56R^2\|\Sigma\|}) \cdot \left[\|\beta_k\|^2 + \sigma_k^2 \cdot \frac{dn}{\lambda_{\min}^{\mathcal{C}}}\right] \\
&\quad + \sigma_k^2 \cdot \left[\frac{2d}{n\lambda_{\min}^{\mathcal{C}}} + \frac{2d^2n}{\lambda_{\min}^{\mathcal{C}}} \exp(-\frac{3n(\lambda_{\min}^{\mathcal{C}})^2}{56R^2\|\Sigma\|})\right] \\
&\leq \frac{2d\sigma_k^2}{n\lambda_{\min}^{\mathcal{C}}} + \exp(-\frac{3n(\lambda_{\min}^{\mathcal{C}})^2}{56R^2\|\Sigma\|}) \cdot \left[2d\|\beta_k\|^2 + \frac{4nd\sigma_k^2}{\lambda_{\min}^{\mathcal{C}}}\right] \\
&= \frac{2d\sigma_k^2}{n\lambda_{\min}^{\mathcal{C}}} + o(n_k^{-2})
\end{aligned}
$$

$\square$

### H.3. Proof of Theorem 5.5

*Proof.* This result is directly from Lemma 5.4 and Theorem 4.1. $\square$

## I. Proof of Appendix

This section gives the proof of lemmas and theorems shown in Appendix, which includes Lemma B.2, Lemma C.1, Lemma C.3, Lemma C.4, Theorem C.5 and Theorem C.6.

### I.1. Proof of Lemma B.2

*Proof.* For the general $\sigma$-subgaussian case, for the right tail with $t > 0$, we have:

$$
\begin{aligned}
\mathbb{E}[\exp(t(X^2 - \mathbb{E}[X^2]))] &= \exp(-t\sigma_X^2)\big(1 + t\mathbb{E}[(X^2)] + \sum_{i=2}^{\infty} \frac{t^i\mathbb{E}[X^{2i}]}{i!}\big) \\
&\leq \exp(-t\sigma_X^2)\big(1 + t\sigma_X^2 + \sum_{i=2}^{\infty} t^i 2^{i+1}\sigma_s^{2i}\big) \\
&= \exp(-t\sigma_X^2)\big(1 + t\sigma_X^2 + \frac{8\sigma^4 t^2}{1 - 2\sigma_s^2 t}\big) \\
&\leq \exp(\frac{8\sigma^4 t^2}{1 - 2\sigma_s^2 t})
\end{aligned}
$$

where the first inequality is based on $\mathbb{E}[X^{2q}] \leq 2q!2^q\sigma^{2q}$ for integer $q \geq 1$, which is Theorem 2.1 in Boucheron et al. (2013), the last inequality is from $1 + x \leq e^x$. For the left tail, with $t > 0$ we have:

$$
\begin{aligned}
\mathbb{E}[\exp(t(\mathbb{E}[X^2] - X^2))] &\leq 1 + t\mathbb{E}[(\mathbb{E}[X^2] - X^2)] + \frac{\exp(t\sigma_X^2) - t\sigma_X^2 - 1}{\sigma_X^4}\mathbb{V}(X^2) \\
&= 1 + \frac{\exp(t\sigma_X^2) - t\sigma_X^2 - 1}{\sigma_X^4}\mathbb{V}(X^2)
\end{aligned}
$$

where the first inequality is from Theorem 2.9 in Boucheron et al. (2013). For $0 < t < 3/\sigma^2 \leq 3/\sigma_X^2$, we have:

$$
\begin{aligned}
\psi_{\mathbb{E}[X^2]-X^2}(t) &\leq \frac{\exp(t\sigma_X^2) - t\sigma_X^2 - 1}{\sigma_X^4}\mathbb{V}(X^2) \\
&\leq \frac{\mathbb{V}(X^2)}{\sigma_X^4} \frac{\sigma_X^4 t^2}{2(1 - t\sigma_X^2/3)} \\
&\leq \frac{8\sigma^4 t^2}{1 - t\sigma^2/3}
\end{aligned}
$$

where the first inequality is from $\log(1 + x) \leq x$, the second inequality is from $e^x - x - 1 \leq x^2/(2 - 2x/3)$ for $0 < x < 3$, the last inequality is from $\mathbb{V}(X^2) \leq \mathbb{E}[X^4] \leq 16\sigma^4$.

For the special case $\sigma_X^2 = \sigma^2$, for the right tail with $t > 0$, we have:

$$
\begin{aligned}
\mathbb{E}[\exp(t(X^2 - \mathbb{E}[X^2]))] &= \frac{\exp(-t\sigma_X^2)}{\sqrt{4\pi t}} \int_{-\infty}^{\infty} \mathbb{E}[\exp(sX)] \exp\left(-\frac{s^2}{4t}\right) ds \\
&\leq \frac{\exp(-t\sigma_X^2)}{\sqrt{4\pi t}} \int_{-\infty}^{\infty} \exp\left(\frac{\sigma_X^2 s^2}{2} - \frac{s^2}{4t}\right) ds \\
&= \frac{\exp(-t\sigma_X^2)}{\sqrt{4\pi t}} \cdot \sqrt{\frac{4\pi t}{1 - 2\sigma_X^2 t}} \\
&\leq \exp\left(\frac{\sigma_X^4 t^2}{1 - 2\sigma_X^2 t}\right)
\end{aligned}
$$

where the first equality is from Hubbard–Stratonovich transformation, the first inequality uses the subgaussian property, the second equality is based on $\int_{-\infty}^{\infty} \exp(-as^2)ds = \sqrt{\pi/a}$ for $a > 0$, the last inequality is by $-\log(1 - u) - u \leq u^2/[2(1 - u)]$ for $0 < u < 1$. For the left tail, the proof is similar as the general case, only in the last step we have $\mathbb{V}(X^2) = \mathbb{E}[X^4] - \sigma_X^4 \leq 2\sigma_X^4$. $\qquad\square$

### I.2. Proof of Lemma C.1

*Proof.* If $\xi_\tau(\delta)$ holds and $\varepsilon_\tau^-(\delta) < \sigma_{\min}^2$, then we have:

$$
\begin{aligned}
R_\infty(\boldsymbol{\lambda}_{\pi_1}) - R_\infty(\boldsymbol{\lambda}^*) &= \frac{1}{T}\left(\max_k \frac{\sigma_k^2}{\lambda_k} - \Sigma_2\right) \\
&= \frac{1}{T}\left(\max_k \frac{\sigma_k^2 \hat{\Sigma}_{2,\tau}}{\hat{\sigma}_{k,\tau}^2} - \Sigma_2\right) \\
&\leq \frac{1}{T}\left(\max_k \frac{\Sigma_2 - \sigma_k^2 + (K-1)\varepsilon_\tau^+}{1 - \varepsilon_\tau^-/\sigma_k^2} - (\Sigma_2 - \sigma_k^2)\right) \\
&= \frac{1}{T}\left(\frac{\Sigma_2 - \sigma_{\min}^2 + (K-1)\varepsilon_\tau^+}{1 - \varepsilon_\tau^-/\sigma_{\min}^2} - (\Sigma_2 - \sigma_{\min}^2)\right) \\
&= \frac{1}{T}\left[(K-1)\varepsilon_\tau^+ + \frac{\varepsilon_\tau^-}{\sigma_{\min}^2}(\Sigma_2 - \sigma_{\min}^2) + (K-1)\varepsilon_\tau^+ \cdot \sum_{n=1}^{\infty}\left(\frac{\varepsilon_\tau^-}{\sigma_{\min}^2}\right)^n + (\Sigma_2 - \sigma_{\min}^2)\sum_{n=2}^{\infty}\left(\frac{\varepsilon_\tau^-}{\sigma_{\min}^2}\right)^n\right] \\
&= \frac{1}{T}\left[(K-1)\varepsilon_\tau^+ + \left(\frac{\Sigma_2}{\sigma_{\min}^2} - 1\right)\varepsilon_\tau^- + \frac{(K-1)\varepsilon_\tau^+\varepsilon_\tau^-}{\sigma_{\min}^2 - \varepsilon_\tau^-} + \frac{(\Sigma_2 - \sigma_{\min}^2)(\varepsilon_\tau^-)^2}{\sigma_{\min}^2(\sigma_{\min}^2 - \varepsilon_\tau^-)}\right] \\
&= \frac{1}{T}\left[(K-1)\varepsilon_\tau^+ + \left(\frac{\Sigma_2}{\sigma_{\min}^2} - 1\right)\varepsilon_\tau^-\right] + o(T^{-3/2})
\end{aligned}
$$

where for the third equality, the maximization is attained when we choose $\hat{\sigma}_{K,\tau}^2 = \sigma_K^2 - \varepsilon_\tau^-(\delta)$ for the lowest-variance arm $K$ and $\hat{\sigma}_{k,\tau}^2 = \sigma_k^2 + \varepsilon_\tau^+(\delta)$ for the other arms, the fourth equality is from $(1 - x)^{-1} = \sum_{n=0}^{\infty} x^n$ for $x < 1$. $\qquad\square$

### I.3. Proof of Lemma C.3

*Proof.* Since $\varepsilon_\tau^- < \sigma_{\min}^2$, assume $\varepsilon_\tau^- \leq \sigma_{\min}^2/m$ for $m > 1$. Based on Lemma C.2, we first bound the term $\sum_{k=1}^{K} \frac{(\lambda_{k,\pi_1} - \lambda_k^*)^2}{\lambda_k^*}$. Then, we control the third-order remainder of Taylor expansion. Firstly, as we have $\sigma_{\min}^2$ by us-

ing a slightly loose upper bound, we have:

$$\sum_{k=1}^{K} \frac{(\lambda_{k,\pi_1} - \lambda_k^*)^2}{\lambda_k^*} = \sum_{k=1}^{K} \frac{[(\hat{\sigma}_{k,\tau}^q - \sigma_k^q) - \lambda_k^*(\hat{\Sigma}_{q,\tau} - \Sigma_q)]^2}{\lambda_k^*(\hat{\Sigma}_{q,\tau})^2}$$

$$\leq \frac{\Sigma_q}{(\hat{\Sigma}_{q,\tau})^2} \sum_{k=1}^{K} \frac{(\hat{\sigma}_{k,\tau}^q - \sigma_k^q)^2}{\sigma_k^q}$$

$$\leq \frac{\Sigma_q}{(\sum_{k=1}^{K}(\sigma_k^2 - \varepsilon_\tau^-)^{q/2})^2} \sum_{k=1}^{K} \frac{[(\sigma_k^2 + \varepsilon_\tau^+)^{q/2} - \sigma_k^q]^2}{\sigma_k^q}$$

$$\leq \frac{q^2(\varepsilon_\tau^+)^2 \Sigma_{q-4}}{4\Sigma_q} \sum_{n=0}^{\infty} (n+1)(\frac{mq\varepsilon_\tau^- \Sigma_{q-2}}{2(m-1)\Sigma_q})^n$$

$$= \frac{q^2(\varepsilon_\tau^+)^2 \Sigma_{q-4}}{4\Sigma_q} + o(T^{-1})$$

where the second inequality is from $(\sigma_k^2 + \varepsilon_\tau^+)^{q/2} - \sigma_k^q \leq q\sigma_k^{q-2}\varepsilon_\tau^+/2$, the third inequality is from $(\sigma_k^2 - \varepsilon_\tau^-)^{q/2} \geq \sigma_k^q - mq\varepsilon_\tau^- \sigma_k^{q-2}/(2m-2)$. Then for the reminder term, first for $||\boldsymbol{\lambda}_{\pi_1} - \boldsymbol{\lambda}^*||_\infty^3$ we can get that:

$$||\boldsymbol{\lambda}_{\pi_1} - \boldsymbol{\lambda}^*||_\infty^3 = \max_k \left| \frac{\hat{\sigma}_k^q}{\hat{\Sigma}_q} - \frac{\sigma_k^q}{\Sigma_q} \right|^3$$

$$\leq \max_k \left( \frac{|\hat{\sigma}_k^q - \sigma_k^q|}{\hat{\Sigma}_q} + \frac{\sigma_k^q|\hat{\Sigma}_q - \Sigma_q|}{\hat{\Sigma}_q \Sigma_q} \right)^3$$

$$\leq \max_k \left[ \frac{\frac{q}{2}(\sigma_k^2 - \varepsilon_\tau^-)^{\frac{q}{2}-1}|\sigma_k^2 - \hat{\sigma}_k^2|}{\sum_{j=1}^{K}(\sigma_j^2 - \varepsilon_\tau^-)^{q/2}} + \frac{\sigma_k^q \varepsilon_\tau^+ \sum_{j=1}^{K} \frac{q}{2}(\sigma_j^2 - \varepsilon_\tau^-)^{\frac{q}{2}-1}}{\sum_{j=1}^{K}(\sigma_j^2 - \varepsilon_\tau^-)^{q/2}\Sigma_q} \right]^3$$

$$\leq \left[ \frac{q(K+1)}{2K\sigma_{\min}^2(1-1/m)} \right]^3 (\varepsilon_\tau^+)^3$$

where the third inequality is from $(\sigma_j^2 - \varepsilon_\tau^-)^{\frac{q}{2}-1} \leq 1$ and $|\sigma_k^2 - \hat{\sigma}_k^2| \leq \varepsilon_\tau^+$. Then for the term $\max_k \frac{\lambda_k^*}{\lambda_{k,\pi_1}}$, we have:

$$\max_k \frac{\lambda_k^*}{\lambda_{k,\pi_1}} = \max_k \frac{\sigma_k^q}{\Sigma_q} / (\frac{\hat{\sigma}_k^q}{\hat{\Sigma}_q})$$

$$\leq \max_k \frac{\sigma_k^q}{\Sigma_q}(1 + (\sigma_k^2 - \varepsilon_\tau^-)^{-q/2} \sum_{j \neq k}(\sigma_j^2 + \varepsilon_\tau^+)^{q/2})$$

$$\leq \max_k \frac{1}{\Sigma_q}(\sigma_k^q + (1-1/m)^{-q/2} \sum_{j \neq k}(\sigma_j^2 + \varepsilon_\tau^+)^{q/2})$$

$$\leq \max_k \frac{1}{\Sigma_q}(\sigma_k^q + (1-1/m)^{-q/2} \sum_{j \neq k}(\sigma_j^q + q\sigma_j^{q-2}\varepsilon_\tau^+/2))$$

$$\leq \max_k \frac{1}{\Sigma_q}(\sigma_k^q + (1-1/m)^{-q/2}(\Sigma_q - \sigma_k^q + q\varepsilon_\tau^+(\Sigma_{q-2} - \sigma_k^{q-2})/2))$$

$$\leq \max_k (1-1/m)^{-q/2}(1 + \frac{q(\Sigma_{q-2} - \sigma_k^{q-2})}{2\Sigma_q}\varepsilon_\tau^+)$$

$$\leq (1-1/m)^{-q/2}(1 + \frac{q\Sigma_{q-2}}{2\Sigma_q}\varepsilon_\tau^+)$$

$$\leq (1-1/m)^{-q/2} + \mathcal{O}(\sqrt{T^{-1}\log T})$$

where the third inequality is from $(\sigma_k^2 + \varepsilon_\tau^+)^{q/2} - \sigma_k^q \le q\sigma_k^{q-2}\varepsilon_\tau^+/2$. Combine them together, we have:

$$
\begin{aligned}
R_p(\boldsymbol{\lambda}_{\pi_1}) - R_p(\boldsymbol{\lambda}^*) &\le \frac{(p+1)R_p(\boldsymbol{\lambda}^*)}{2}\sum_{k=1}^{K}\frac{(\lambda_{k,\pi_1} - \lambda_k^*)^2}{\lambda_k^*} + \frac{7(p+2)^2}{\lambda_{\min}^* T}\max_k(\frac{\lambda_k^*}{\lambda_{k,\pi_1}})^{3p+3}\|\boldsymbol{\lambda}_{\pi_1} - \boldsymbol{\lambda}^*\|_\infty^3 \\
&\le \frac{(p+1)(\Sigma_q)^{2/q}}{2T}\left[\frac{q^2(\varepsilon_\tau^+)^2\Sigma_{q-4}}{4\Sigma_q}\right] + o(T^{-2}) \\
&= \frac{p^2(\Sigma_q)^{1/p}\Sigma_{q-4}}{2(p+1)T}(\varepsilon_\tau^+)^2 + o(T^{-2})
\end{aligned}
$$

$\square$

## I.4. Proof of Lemma C.4

*Proof.* Based on $\xi_T(\delta)$, we have:

$$
\sigma_k^2 - \varepsilon_n^- - \varepsilon_n^+ \le \text{LCB}_{k,n} \le \sigma_k^2 \le \text{UCB}_{k,n} \le \sigma_k^2 + \varepsilon_n^- + \varepsilon_n^+.
$$

Assume $e_{\tau_k} = \varepsilon_{\tau_k}^+ + \varepsilon_{\tau_k}^- \le \sigma_k^2/m_{k,\tau_k}$ for $m_{k,\tau_k} > 1$, and we can get that:

$$
\begin{aligned}
\tau_k &\ge \alpha_k T \\
&\ge \frac{(\sigma_k^2 - e_{\tau_k})^{q/2}}{(\sigma_k^2 - e_{\tau_k})^{q/2} + \sum_{j\neq k}(\sigma_j^2 + e_{\tau_j})^{q/2}}T \\
&\ge \frac{(\sigma_k^2 - \sigma_k^2/m_{k,\tau_k})^{q/2}}{(\sigma_k^2 - \sigma_k^2/m_{k,\tau_k})^{q/2} + \sum_{j\neq k}(\sigma_j^2 + \sigma_j^2/m_{j,\tau_j})^{q/2}}T \\
&\ge \min_{j\neq k}\left[\frac{1 - 1/m_{k,\tau_k}}{1 + 1/m_{j,\tau_j}}\right]^{q/2}\lambda_k^* T
\end{aligned}
$$

We can get $\tau_k = \Omega(T)$, and since $\tau_k \le \lambda_k^* T$, we have $\tau_k = \Theta(T)$. Then we need to derive the relationship between $m_{k,\tau_k}$ and $T$ for all $k = 1, \cdots, K$:

$$
\begin{aligned}
m_{k,\tau_k} &= \frac{\sigma_k^2}{\varepsilon_{\tau_k}^+ + \varepsilon_{\tau_k}^-} \\
&= \frac{\sigma_k^2}{\sigma^2}\left(8\sqrt{2}\frac{1 + \sqrt{\tau_k - 1}}{\sqrt{\tau_k - 1}}\sqrt{\frac{\alpha_0\log T}{\tau_k}} + \frac{19\alpha_0\log T}{3\tau_k}\right)^{-1} \\
&\ge \frac{\sigma_k^2}{\sigma^2}\frac{1}{8\sqrt{2}\left(1 + \frac{1}{\sqrt{\tau_k - 1}}\right)\sqrt{\frac{\alpha_0\log T}{\tau_k}}}\left[1 - \frac{\frac{19\,\alpha_0\log T}{3\tau_k}}{8\sqrt{2}\left(1 + \frac{1}{\sqrt{\tau_k - 1}}\right)\sqrt{\frac{\alpha_0\log T}{\tau_k}}}\right] \\
&= \frac{\sigma_k^2}{\sigma^2}\frac{1}{8\sqrt{2}}\frac{1}{\left(1 + \frac{1}{\sqrt{\tau_k - 1}}\right)}\sqrt{\frac{\tau_k}{\alpha_0\log T}}\left[1 - \frac{19}{24\sqrt{2}}\frac{\sqrt{\alpha_0\log T/\tau_k}}{1 + \frac{1}{\sqrt{\tau_k - 1}}}\right] \\
&= \frac{\sigma_k^2}{\sigma^2}\left[\frac{1}{8\sqrt{2}}\frac{1}{1 + \frac{1}{\sqrt{\tau_k - 1}}}\sqrt{\frac{\tau_k}{\alpha_0\log T}} - \frac{19}{384}\frac{1}{\left(1 + \frac{1}{\sqrt{\tau_k - 1}}\right)^2}\right] \\
&\ge \frac{\sigma_k^2}{\sigma^2}\left[\frac{1}{8\sqrt{2}}\frac{1}{1 + \frac{1}{\sqrt{\tau_k - 1}}}\sqrt{\frac{\tau_k}{\alpha_0\log T}} - \frac{19}{384}\right] \\
&= \Omega\left(\sqrt{\frac{T}{\log T}}\right)
\end{aligned}
$$

where $\alpha_0 = 2$ when $p = \infty$ and $\alpha_0 = 5/2$ when $p < \infty$, the first inequality is from $(A + B)^{-1} \geq A^{-1}(1 - B/A)$ for $0 \leq B \leq A$. Also, it is easy to verify that

$$m_{k,\tau_k} \leq \frac{\sigma_k^2}{\sigma^2} \frac{1}{8\sqrt{2}} \sqrt{\frac{\tau_k}{\alpha_0 \log T}} = \mathcal{O}\left(\sqrt{\frac{T}{\log T}}\right).$$

Then we have $m_{k,\tau_k} = \Theta(\sqrt{\frac{T}{\log T}})$. Let $\underline{m_k} = \min_{j \neq k} m_{j,\tau_j}$, then $\underline{m_k} = \Theta(\sqrt{\frac{T}{\log T}})$, then we can get:

$$\begin{aligned}
\alpha_k &= \min_{j \neq k} \left[\frac{1 - 1/m_{k,\tau_k}}{1 + 1/m_{j,\tau_j}}\right]^{q/2} \lambda_k^* \\
&= \left(\frac{1 - 1/m_{k,\tau_k}}{1 + 1/\underline{m_k}}\right)^{q/2} \lambda_k^* \\
&= \lambda_k^* \left[1 - \frac{q}{2}(1/m_{k,\tau_k} + 1/\underline{m_k}) + \mathcal{O}(T^{-1} \log T)\right].
\end{aligned}$$

where the third equality is from Taylor expansion of $(1 - x)^a$ and $(1 + x)^{-a}$. Then assume

$$m_{k,\tau_k}, \underline{m_k} \in [c_1 \sqrt{T/\log T}, c_2 \sqrt{T/\log T}],$$

then we can get:

$$\lambda_k^* \left[1 - \frac{q}{c_2}\sqrt{T^{-1} \log T}\right] \leq \alpha_k \leq \lambda_k^* \left[1 - \frac{q}{c_1}\sqrt{T^{-1} \log T} + \mathcal{O}(T^{-1} \log T)\right].$$

This shows that $\alpha_k = \lambda_k^*(1 - \Theta(\sqrt{T^{-1} \log T}))$. $\qquad\square$

### I.5. Proof of Theorem C.5

*Proof.* For $p = \infty$, in the event $\xi_\tau(\delta)$, we would have:

$$\begin{aligned}
\max_k \frac{\sigma_k^2}{\lambda_{k,\pi_1}} - \sum_{j=1}^K \sigma_j^2 &= \max_k \frac{\sigma_k^2 \hat{\Sigma}_{2,\tau}}{\hat{\sigma}_{k,\tau}^2} - \Sigma_2 \\
&\leq \max_k \sum_{j \neq k} (\sigma_j^2 + \varepsilon_{j,\tau}^+)\left(1 - \frac{\varepsilon_{k,\tau}^-}{\sigma_k^2}\right)^{-1} - (\Sigma_2 - \sigma_k^2) \\
&= \left(\frac{1 + s_\tau^+}{1 - s_\tau^-} - 1\right)(\Sigma_2 - \sigma_{\min}^2) \\
&\leq (s_\tau^+ + s_\tau^-)(\Sigma_2 - \sigma_{\min}^2) + o(T^{-1/2})
\end{aligned}$$

Then we have:

$$\begin{aligned}
\mathbb{E}\left[R_\infty(\boldsymbol{n}_{\pi_1}) - R_\infty(\boldsymbol{n}^*)\right] &= \mathbb{E}\left[R_\infty(\boldsymbol{\lambda}) - R_\infty(\boldsymbol{\lambda}^*)\right] \\
&\leq \mathbb{E}\left[R_\infty(\boldsymbol{\lambda}) - R_\infty(\boldsymbol{\lambda}^*)|\xi_\tau(\delta)\right] + 2KT^{-1}\mathbb{E}\left[R_\infty(\boldsymbol{\lambda}) - R_\infty(\boldsymbol{\lambda}^*)|\xi_\tau^c(\delta)\right] \\
&\leq \frac{1}{T}\left[(s_\tau^+ + s_\tau^-)(\Sigma_2 - \sigma_{\min}^2)\right] + o(T^{-3/2}) \\
&\leq \frac{4(\Sigma_2 - \sigma_{\min}^2)}{T}\sqrt{\frac{\log T}{\tau}} + o(T^{-3/2}) \\
&\leq 4\lambda^{-1/2}(\Sigma_2 - \sigma_{\min}^2)T^{-3/2}\sqrt{\log T} + o(T^{-3/2})
\end{aligned}$$

When $p$ is finite, in the event $\xi_\tau(\delta)$, we would have:

$$\sum_{k=1}^{K} \frac{(\lambda_{k,\pi_1} - \lambda_k^*)^2}{\lambda_k^*} \leq \frac{\Sigma_q}{(\sum_{k=1}^{K}(\sigma_k^2 - \varepsilon_{k,\tau}^-)^{q/2})^2} \sum_{k=1}^{K} \frac{[(\sigma_k^2 + \varepsilon_{k,\tau}^+)^{q/2} - \sigma_k^q]^2}{\sigma_k^q}$$

$$= \frac{[(1 + s_\tau^+)^{q/2} - 1]^2}{(1 - s_\tau^-)^q}$$

$$\leq \frac{q^2(s_\tau^+)^2}{4(1 - s_\tau^-)^q}$$

$$\leq \frac{q^2(s_\tau^+)^2}{4} + o(T^{-1})$$

Then we have:

$$\mathbb{E}\left[R_p(\boldsymbol{n}_{\pi_1}) - R_p(\boldsymbol{n}^*)\right] = \mathbb{E}\left[R_p(\boldsymbol{\lambda}) - R_p(\boldsymbol{\lambda}^*)\right]$$

$$\leq \mathbb{E}\left[R_p(\boldsymbol{\lambda}) - R_p(\boldsymbol{\lambda}^*)|\xi_\tau(\delta)\right] + 2KT^{-3/2}\mathbb{E}\left[R_p(\boldsymbol{\lambda}) - R_p(\boldsymbol{\lambda}^*)|\xi_\tau^c(\delta)\right]$$

$$\leq \frac{p^2(\Sigma_q)^{2/q}}{2(p+1)T}(s_\tau^+)^2 + o(T^{-2})$$

$$= \frac{3p^2(\Sigma_q)^{2/q}}{\lambda(p+1)}T^{-2}\log T + o(T^{-2})$$

$\square$

### I.6. Proof of Theorem C.6

*Proof.* For $p = \infty$, in the event $\xi_T(\delta)$, we would have:

$$\max_k \frac{\sigma_k^2}{\lambda_{k,\pi_2}} - \Sigma_2 \leq \max_k \sum_{j \neq k}\left(\sigma_j^2 + \varepsilon_{\tau_j}^+\right)\left(1 + \sum_{n=1}^{\infty}\left(\frac{\varepsilon_{\tau_k}^-}{\sigma_k^2}\right)^n\right) - (\Sigma_2 - \sigma_k^2)$$

$$= (\Sigma_2 - \sigma_{\min}^2)s_{\tau_K}^- + \sum_{j \neq K} s_{\tau_j}^+ \sigma_j^2 + \mathcal{O}(T^{-1}\log T)$$

$$= 2(\Sigma_2 - \sigma_{\min}^2)\sqrt{\frac{\log(1/\delta)}{\tau_K}} + 2\sum_{j \neq K}\sigma_j^2\sqrt{\frac{\log(1/\delta)}{\tau_j}} + \mathcal{O}(T^{-1}\log T)$$

$$\leq \sqrt{8T^{-1}\log T}\frac{\Sigma_2 - 2\sigma_{\min}^2}{\sqrt{\alpha_K}} + \sqrt{8T^{-1}\log T}\sum_{k=1}^{K}\frac{\sigma_k^2}{\sqrt{\alpha_k}} + \mathcal{O}(T^{-1}\log T)$$

$$\leq \sqrt{8T^{-1}\log T}\frac{\Sigma_2 - 2\sigma_{\min}^2}{\sqrt{\alpha_K}} + \sqrt{8T^{-1}\log T}\sum_{k=1}^{K}\frac{\sigma_k^2}{\sqrt{\alpha_k}} + \mathcal{O}(T^{-1}\log T)$$

$$\leq \sqrt{8T^{-1}\log T}\left[\frac{\sqrt{\Sigma_2}(\Sigma_2 - 2\sigma_{\min}^2)}{\sigma_{\min}} + \sqrt{\Sigma_2}\Sigma_1\right] + \mathcal{O}(T^{-1}\log T)$$

Then we can get that:

$$\mathbb{E}\left[R_\infty(\boldsymbol{n}_{\pi_2}) - R_\infty(\boldsymbol{n}^*)\right] = \mathbb{E}\left[R_\infty(\boldsymbol{\lambda}) - R_\infty(\boldsymbol{\lambda}^*)\right]$$

$$\leq \mathbb{E}\left[R_\infty(\boldsymbol{\lambda}) - R_\infty(\boldsymbol{\lambda}^*)|\xi_T(\delta)\right] + 2T^{-1}\mathbb{E}\left[R_\infty(\boldsymbol{\lambda}) - R_\infty(\boldsymbol{\lambda}^*)|\xi_T^c(\delta)\right]$$

$$\leq 2\sqrt{2}\left[\frac{\sqrt{\Sigma_2}(\Sigma_2 - 2\sigma_{\min}^2)}{\sigma_{\min}} + \sqrt{\Sigma_2}\Sigma_1\right]T^{-3/2}\sqrt{\log T} + o(T^{-3/2})$$

When $p$ is finite, in the event $\xi_T(\delta)$, we would have:

$$
\begin{aligned}
\sum_{k=1}^{K} \frac{(\lambda_{k,\pi_2} - \lambda_k^*)^2}{\lambda_k^*} &\leq \frac{\Sigma_q}{(\sum_{k=1}^{K}(\sigma_k^2 - \varepsilon_{\tau_k}^-)^{q/2})^2} \sum_{k=1}^{K} \frac{[(\sigma_k^2 + \varepsilon_{\tau_k}^+)^{q/2} - \sigma_k^q]^2}{\sigma_k^q} \\
&= \frac{\sum_{k=1}^{K} \sigma_k^q[(1 + s_{\tau_k}^+)^{q/2} - 1]^2}{[\sum_{k=1}^{K} \sigma_k^q(1 - s_{\tau_k}^-)^q]^2} \Sigma_q \\
&\leq \frac{q^2 \sum_{k=1}^{K} \sigma_k^q(s_{\tau_k}^+)^2}{4\Sigma_q(1 - s_{\tau_{\min}}^-)^{2q}} \\
&\leq \frac{5q^2 \sum_{k=1}^{K} \sigma_k^q/\alpha_k}{2\Sigma_q T} \log T + o(T^{-1}) \\
&\leq \frac{5}{2} K q^2 T^{-1} \log T + o(T^{-1})
\end{aligned}
$$

Then we have:

$$
\begin{aligned}
\mathbb{E}\left[R_\infty(\boldsymbol{n}_{\pi_2}) - R_\infty(\boldsymbol{n}^*)\right] &= \mathbb{E}\left[R_\infty(\boldsymbol{\lambda}) - R_\infty(\boldsymbol{\lambda}^*)\right] \\
&\leq \mathbb{E}\left[R_\infty(\boldsymbol{\lambda}) - R_\infty(\boldsymbol{\lambda}^*)|\xi_T(\delta)\right] + 2T^{-3/2}\mathbb{E}\left[R_\infty(\boldsymbol{\lambda}) - R_\infty(\boldsymbol{\lambda}^*)|\xi_T^c(\delta)\right] \\
&\leq \frac{5}{4}(p+1)Kq^2(\Sigma_q)^{2/q}T^{-2}\log T + o(T^{-2}) \\
&= \frac{5Kp^2(\Sigma_q)^{2/q}}{p+1}T^{-2}\log T + o(T^{-2})
\end{aligned}
$$

$\square$

