# OpenReview forum: "Exploration-free Algorithms for Multi-group Mean Estimation"
_ICML.cc/2026/Conference — ICML 2026 regular_

### Official Review · Reviewer_9JwP · 2026-02-24

**Soundness:** 3
**Presentation:** 3
**Significance:** 2
**Originality:** 2
**Overall Recommendation:** 2
**Confidence:** 4

**Summary:**

The paper considers the problem of experimental design in the stochastic multi-armed bandit problem. Instead of the usual goal of minimizing the regret, they consider the goal of estimating all the means well -- and in particular, aiming to ensure that the \ell_p norm of the MSE vector is small. If the variances of each arm were known there is a simple easy-to-derive closed form solution to how many samples should be allocated to each arm, so in essence their problem boils down to estimating the variance of each arm. They work under a sub-Gaussian assumption for the rewards, and improve existing variance concentration bounds, and build on this to design simple adaptive and near-optimal schemes.

**Compliance With Llm Reviewing Policy:**

Affirmed.

**Final Justification:**

I remain confident in my assessment that this paper is below the acceptance threshold.

I appreciate the authors' perspective that a (technically) simple paper can have virtues, and be a valuable contribution. I do not agree that this particular manuscript meets any reasonable criteria to warrant acceptance at the premier machine learning conference.

**Key Questions For Authors:**

I am reasonably confident in my assessment. If I am making some gross oversimplification please let me know and I will update my score/review.

**Limitations:**

Yes.

**Strengths And Weaknesses:**

Strengths:

The authors have a clear setup and goal, and derive near-optimal algorithms to achieve their stated goal.

Weaknesses:

1. To me the main weakness is that the paper, as a mainly theoretical contribution, does not reach the level of technical sophistication required for a competitive ICML submission. One of their main technical contributions is to observe that the Hanson-Wright inequality could be used to provide a variance concentration inequality for sub-Gaussians (but of course, this fact is well-known see for instance the work of Dicker-Erdogdu).

2. Secondarily, all of the insights and methodology of the paper are very brittle to their exact setup. The sub-Gaussian assumption is strong (and is probably easy to relax, using some ideas from the robust estimation literature). The other insights (of requiring \Theta(T) allocation to each arm) are all direct consequences of the uniform estimation goal.

3. Finally, it would strengthen the paper a lot to have a clearer discussion of the practical applications of this methodology. While they briefly mention personalization and experimental design, it would be helpful to showcase real-world applications which follow something like their theoretical setup.

---

> ### Author Rebuttal · Authors · 2026-03-31
>
> We thank the reviewer for all the comments. Here are our responses to the questions.
>
> # Theoretical Contribution
> Our goal is not technical complexity for its own sake, but to clarify a key structural property of the problem and build theory and algorithms around it. Prior work [1,2] did not fully exploit that the oracle allocation gives each arm $\Theta(T)$ samples, and continued to rely on a suboptimal variance concentration bound. Our paper makes this structure explicit, explains its algorithmic consequences, and develops a simple but effective allocation rule and sharper analysis.
>
> ## Subgaussian Variance Concentration
> The variance bound used in prior work (Theorem B.5) is widely cited and used, but is statistically suboptimal as sample variance is a U-statistic and should exhibit CLT-type scaling. We agree that extending the analysis to broader classes such as $\alpha$-subexponential distributions [3] would be valuable. But we want to emphasize some points:
>
> As matrix $A$ has all diagonal entries $1/n$ and the off-diagonal entries $-\frac{1}{n(n-1)}$, it allows a refinement beyond the generic Hanson-Wright bound to derive a sharper but still explicit bound; see Appendix B.2.
>
> Secondly, while there is a line of literature on refined Hanson-Wright inequality, many such results involve unspecified constants or auxiliary parameters that make them difficult to use directly in algorithm design. In contrast, one of our goals is to provide a sharper bound that remains explicit and algorithmically usable.
>
> Lastly, even $\sigma$-subgaussian case can be difficult for moderate $T$. As informative lower confidence bounds require enough samples, which grows with $\sigma$, the general subgaussian often needs a large $T$ in simulation study for a large $K$. This partly explains why prior work [1,2] focused on simple Gaussian two-arm case, and motivates our strictly subgaussian setting.
>
> ## Algorithm Design
> Prior work mainly adopts UCB-style designs, which are natural in traditional bandits. Our main point is that such designs are not fully aligned with the objective, as the oracle pulls every arm $\Theta(T)$ times. This structure motivates our exploration-free perspective, reformulates the problem in terms of estimating $\lambda_k$, simplifies the analysis, and enables the reactivation step in the adaptive algorithm, which keeps $\tau_k$ close to $n_k^*$ without exceeding it. It also allows batched updates, which are computationally more efficient than per-round UCB updates.
>
> While this property may appear natural in hindsight, we believe an important contribution of our paper is to make this $\Theta(T)$ allocation structure explicit and to use it systematically as the basis for improved algorithm design and analysis, which prior work did not do.
>
> On the empirical side, since worst-case analysis does not fully reflect the practical advantage of an algorithm, we include direct comparisons with UCB-based methods in Appendix D.2 (Figure 5 and Table 1). In particular, the gap between “Ours with GSG” and “UCB refined” shows that the huge improvement comes not only from the refined concentration bound, but also from the exploration-free design.
>
> # Practical Applications of the Methodology
> We agree that the practical relevance should be stated more clearly. Our formulation appears naturally in experimental design, especially Neyman allocation and adaptive A/B testing for estimating the average treatment effect (ATE). Following [4], let $$\tau = \mathbb E[Y(1)-Y(0)], $$
> where $Y(1)$ and $Y(0)$ are the potential outcomes under treatment and control, and let $W_t$ denote the assignment for unit $t$. A standard estimator is $$\hat\tau = \frac{1}{n(1)}\sum_{t:W_t=1}Y_t - \frac{1}{n(0)}\sum_{t:W_t=0}Y_t.$$
> Under standard randomized designs, $\hat\tau$ is unbiased and MSE is$$\mathbb E[(\hat\tau-\tau)^2]=\operatorname{Var}(\hat\tau)=\frac{\sigma^2(1)}{n(1)}+\frac{\sigma^2(0)}{n(0)}.$$
> Thus, ATE estimation is essentially a two-group mean estimation problem ($p=1$ norm), where the key design question is how to allocate samples across groups to minimize estimation error. This is closely aligned with our objective. We refer the reviewer to [4,5] for background on adaptive Neyman allocation, and to [6,7] for representative applications in social science.
>
> We look forward to further discussions in the following week.
>
> [1] Aznag, A. et al. An active learning framework for multi-group mean estimation.
>
> [2] Carpentier, A. et al. Upper-confidence-bound algorithms for active learning in multi-armed bandits.
>
> [3] Götze, F. et al. Concentration inequalities for polynomials in α-sub-exponential random variables.
>
> [4] Zhao, J. Adaptive neyman allocation.
>
> [5] Li, J. et al. Optimal adaptive experimental design for estimating treatment effect.
>
> [6] Blackwell, M. et al. Batch adaptive designs to improve efficiency in social science experiments.
>
> [7] Xu, J. et al. MUSE: Multi-Treatment Experiment Design for Winner Selection and Effect Estimation.

---

> > ### Author Rebuttal · Reviewer_9JwP · 2026-04-04
> >
> > I appreciated the rebuttal. At the very least the ATE/Neyman allocation example should feature more prominently in the paper.
> >
> > I am still underwhelmed by the actual contributions of the paper (my "Weakness 1") and the rebuttal doesn't change my opinion significantly on that front. To me, the paper considers a simple idealized setup/goal and provides a simple resolution. It is to me not at the level of an interesting ICML submission.

---

> > > ### Author Response · Authors · 2026-04-05
> > >
> > > We thank the reviewer for the follow-up. We don't want to further bother the reviewer for additional discussions. But we'd like to take the chance to share more of our thoughts about the contribution of the paper.
> > >
> > > For the technical contribution of the paper, it has been discussed above, and we don't want to reiterate here. But we want to detach from the technical details, and make a few general comments:
> > > - There have been a wide range of papers from different communities working on the topic. Suppose this is a "simple" setup, it then should be more important to clarify the existing suboptimal designs/methodologies, and to provide the most natural (and probably simple) solution/algorithm (as ours) to the problem. Specifically, given its "simple" setup, there are many papers from top conf./journals still using sub-optimal/unnecessary designs such as UCB or exploration-based algorithms; while we don't want to chase after the reasons for these sub-optimal designs, should it be more important to make a clarification (to the communities) as our paper did? This was indeed the reason for us to initiate the project at the first place.
> > > - We have worked on papers with more involved and complicated math. As authors and reviewers ourselves, we believe the technical contribution of a paper should be evaluated based on insights and impacts that a technical analysis generates, but not based on the difficulty and involvedness of math. We, as reviewers and researchers ourselves, will be the cheerleader for a simple proof if it resolves some long-standing confusion we had or generates interesting insights. Otherwise, the whole community will collapse into a game of using ChatGPT to write tedious proof/analysis with length of tens of pages.
> > >
> > > We just want to genuinely share our thoughts on the matter, and we thank the reviewer again for the time spent reading and discussing our paper.

---

### Official Review · Reviewer_CBwR · 2026-03-08

**Soundness:** 3
**Presentation:** 2
**Significance:** 3
**Originality:** 3
**Overall Recommendation:** 4
**Confidence:** 2

**Summary:**

This paper studies the problem of multi-group mean estimation in the classical multi-armed bandit setting. The objective is to design an algorithm that sequentially selects an arm at each time step to ensure that the estimated mean of each arm achieves low variance. The authors first establish a concentration inequality for the variance of sub-Gaussian distributions. Building on this result, they propose both adaptive and non-adaptive algorithms and provide theoretical regret bounds for these methods.

**Compliance With Llm Reviewing Policy:**

Affirmed.

**Final Justification:**

Overall, the authors have addressed my concerns, and I keep my current evaluation towards this paper..

**Key Questions For Authors:**

1. What are the technical contributions of this work compared to existing studies? In particular, it would be helpful to include an overview that explains the structure and flow of the analysis presented in the paper.

2. Could the authors provide further clarification on Theorem 4.1? For example, it would be useful to explain the order of the bound with respect to $T$, and to illustrate how the bound behaves in certain special cases to help readers better understand its implications.

**Strengths And Weaknesses:**

### Strengths

1. The problem of estimating the mean value for each group while minimizing variance in the multi-armed bandit setting is important and well-motivated.
2. The paper provides a solid theoretical analysis and offers a comprehensive study of the problem.

### Weaknesses

1. The writing is sometimes difficult to follow. In particular, the analysis in the appendix is not well organized, and the explanations of the results and the structure of the analysis could be clearer.
2. The paper does not include an impact statement, which is required by the ICML author guidelines.
3. The technical contributions of the analysis are not clearly articulated. Much of the proof in the appendix appears to rely heavily on existing work, making it difficult to identify the novel technical contributions of the paper.

---

> ### Author Rebuttal · Authors · 2026-03-31
>
> We thank the reviewer for appreciating our work and for raising several interesting questions.
>
>
> # Writing and Impact Statement
> We will revise the presentation to make the structure of the paper substantially clearer. The omission of the impact statement was unintentional, and we will include it in the revised version.
>
> # Technical Contributions of the Analysis
> The reviewer mentioned that “much of the proof in the appendix appears to rely heavily on existing work.” We would like to clarify that we have intentionally cited all prior tools and proof frameworks that our analysis builds upon. To the best of our knowledge, the main prior ingredients used in our theoretical analysis are:
>
> 1.Proposition 2.1 is the standard oracle allocation result.
>
> 2.For the Hanson-Wright inequality, we build on the proof framework discussed in Epperly’s blog [1]. However,we refine the argument for the specific structure of the sample variance and obtain a sharper bound; see Appendix B.2.
>
> 3.For subgamma argument, we use the general proof technique from [2]. That reference only discusses the Gaussian case (Example 2.12), whereas we extend the argument to the subgaussian case.
>
> 4.For the regret analysis when $p<\infty$, we use the Lemma 4 in [3] to upper bound the $p$-norm regret term by Taylor expansion, but the remainder of the proof is specific to our setting.
>
> Our intention was to be transparent about every prior ingredient we use, if the reviewer has additional references they believe are more directly overlapping with our proofs, we would appreciate the reviewer pointing them out.
>
> # Comparison to Existing Works
> We compare our results with existing work in Appendix B and Appendix C. Here we would summarize more clearly what is new in our analysis:
>
> Variance concentration: A frequently used approach in prior work is Theorem B.5, leading to a bound of the form $O(\frac{\log(1/\delta)}{\sqrt{n}})$. However, the sample variance is a U-statistic and satisfies a CLT. We use the Hanson-Wright inequality to derive a tighter variance concentration result. By “tighter,” we mean not only that the rate matches the CLT-type scaling as $O(\sqrt{\frac{\log(1/\delta)}{n}})$, but also the constants improve the generic Hanson-Wright result; see Appendix B.2.
>
> Algorithm Design: Prior work [3,4] used UCB-style algorithms, which are widely used in bandit prblem. Our point is that in the present problem such designs are not fully aligned with the objective, because the oracle allocation pulls every arm $\Theta(T)$ times.
>
> This $\Theta(T)$ structure is important in several ways. It motivates the exploration-free perspective, allows the problem to be reformulated in terms of estimating $\lambda_k$. It also enables the reactivation idea in our adaptive algorithm, which keeps $\tau_k$ close to $n_k^*$ without exceeding it. In addition, the same structure allows batched updates, which are computationally more efficient than per-round UCB-style updates. While this property may appear natural in hindsight, we believe an important contribution of our paper is to make this $\Theta(T)$-allocation structure explicit and to use it systematically as the basis for improved algorithm design and analysis, which prior work did not do.
>
> Contextual Case: Previous papers [5,6] studied related contextual settings either through matrix-norm-based guarantees or under more restrictive Gaussian assumptions. In contrast, we consider a more general framework with K linear models under subgaussian noise, and we measure performance in Euclidean norm.
>
> # Further Clarification on Theorem 4.1
> We do provide a comparison in Appendix C.5. Our result gives regret of order $O(T^{-1.5}\sqrt{\log T})$ for $p=\infty$, and $O(T^{-2}\log T)$ for finite $p$, which improves over the previous upper bounds as $O(T^{-1.5}\log^{2} T)$ for $p=\infty$, and $O(T^{-2}\log^{2} T)$ for finite $p$.
>
> For these bounds, we don’t think it has too many specific implications since these upper bounds are from worst-case analysis, one implication is that our bounds remove the dependence on $K$ and involve substantially smaller constants. We would encourage the reviewer to look at the strictly subgaussian setting in Appendix C.3, where the bound becomes much sharper, as well as the symmetric Beta example in Appendix D.3, which illustrates a substantial gap between the two theoretical bounds and helps make the difference more concrete in practice.
>
> We are happy to have further discussions in the following week.
>
> [1] Epperly, E. Note to self: Hanson-wright inequality.
>
> [2] Boucheron, S. et al. Concentration Inequalities: A Nonasymptotic Theory of Independence.
>
> [3] Aznag, A. et al. An active learning framework for multi-group mean estimation.
>
> [4] Carpentier, A. et al. Upper-confidence-bound algorithms for active learning in multi-armed bandits.
>
> [5] Riquelme, C et al. Active learning for accurate estimation of linear models.
>
> [6] Fontaine, X. et al. Online a-optimal design and active linear regression.

---

> > ### Author Rebuttal · Reviewer_CBwR · 2026-04-04
> >
> > I thank the authors for their response! I will keep my current score.

---

### Official Review · Reviewer_8h7i · 2026-03-11

**Soundness:** 3
**Presentation:** 4
**Significance:** 3
**Originality:** 3
**Overall Recommendation:** 5
**Confidence:** 3

**Summary:**

The paper studies the problem of allocating a finite sampling budget across multiple groups in order to estimate all group means accurately. The authors consider a setting where each group has unknown variance and the objective is to minimize the aggregate estimation error across groups. They show that when variances are known, the optimal allocation assigns samples proportionally to powers of the variances, and they design an adaptive algorithm that approximate this optimal allocation when variances are unknown. The then derived theoretical results which upper bounds the excess risk of the proposed method for mean estimation.

**Compliance With Llm Reviewing Policy:**

Affirmed.

**Final Justification:**

the authors have successfully addressed my main concerns, i have hence updated my scores to reflect the theoretical contribution.

**Key Questions For Authors:**

Do the authors have any corresponding lower bounds for this problem that characterize the minimax rate of the excess error?

**Limitations:**

Yes

**Strengths And Weaknesses:**

Strengths:

1. Problem is cleanly formulated and explored, the theoretical development is sound.
2. The proposed adaptive algorithm is conceptually simple to understand and easy to implement.
3. Explicit finite sample theoretical guarantees are derived, and the rates match what one would heuristically believe to be the optimal rate.

Weaknesses:

1. The paper provides upper bounds on the excess estimation error relative to the oracle allocation, but does not establish corresponding lower bounds. Without such minimax lower bounds, the claim that the algorithms achieve near-optimal performance is somewhat heuristic.

2. The exploration-free interpretation is somewhat debatable. The paper emphasizes that the proposed algorithms avoid explicit exploration strategies. However, the algorithms still rely on estimating the variances of each group in order to determine the appropriate allocation. From a learning perspective, this initial estimation phase plays a role similar to exploration. I understand that the authors argue that these samples collected are still contributing towards variance estimate. However, this interpretation arguably holds for any algorithm in this setting, since every sample ultimately contributes to the mean estimates. As such, lack of exploration may reflect a property of the problem formulation rather than a feature of the proposed algorithms that the authors want to emphasize on.

3. The particular problem framed here (estimating multiple means) is less commonly encountered in practical sequential decision-making problems compared to traditional bandit formulations. As a result, the practical impact of the problem formulation may be somewhat limited relative to more standard bandit settings.

---

> ### Author Rebuttal · Authors · 2026-03-31
>
> We thank the reviewer for the positive feedback on our work, and for bringing up the question of the lower bound.
>
> # Minimax Lower Bound
>
> We thank the reviewer for raising this point. Prior work [1] already established lower bounds using Le Cam’s two-point method in the two-Gaussian-arm setting, showing rates of $\Omega(T^{-2})$ for finite $p$ and $\Omega(T^{-1.5})$ for $p=\infty$. We believe these are the natural benchmark rates. In our case, we can recover the same rates by considering two arms with sufficiently close variances. Extending such a construction to a general $K$-arm setting seems possible, but in our view it would mainly broaden the statement rather than provide new insight. In the revision, we will cite these lower bounds explicitly and clarify their relationship to our upper-bound analysis.
>
> # Exploration-free Interpretation
>
> We thank the reviewer for this comment. By “exploration-free”, we mean that on the high-probability good event used in our analysis, the total number of samples collected up to the end of Phase 2 ($\tau_k$) would not exceed the oracle allocation ($n_k^*$) for any arm. This property is not satisfied by previous algorithms [1,2]. We agree that the algorithm design is informed by the structure of the problem, but this is precisely the point we want to emphasize: once one recognizes that the oracle solution allocates $\Theta(T)$ samples to every arm, the traditional UCB-based algorithm is no longer well aligned with the objective. Besides, our design simplifies the analysis by reducing the problem to estimating $\lambda_k$​, and it also avoids additional regret caused by exceeding the oracle budget of an arm, since such extra pulls cannot be revoked afterward.
>
> We also respectfully encourage the reviewer to consult our empirical comparison with UCB-based methods (Figure 5 and Table 1 in Appendix D.2), especially the comparison between “Ours with GSG” and “UCB refined.” The observed improvement goes beyond the gain from the refined concentration inequality alone and also reflects the benefit of the exploration-free allocation design itself, which is difficult to fully capture through worst-case theory alone.
>
> # Practical Impact of the Problem Formulation
>
> We thank the reviewer for raising the question of practical relevance. This formulation is common in experimental design, especially in Neyman allocation and adaptive A/B testing for estimating the average treatment effect (ATE) (following the notation in [3]):
> $$ \tau = \mathbb{E}[Y(1)-Y(0)], $$
> where $Y(1)$ and $Y(0)$ denote the potential outcomes under treatment and control respectively. Let $W_t$ represent the treatment assignment that unit $t$ receives. A standard estimator is the difference-in-means estimator:
> $$\hat{\tau} = \frac{1}{n(1)} \sum_{t: W_t = 1} Y_t - \frac{1}{n(0)} \sum_{t: W_t = 0} Y_t. $$
>
> Under standard randomized designs, $\hat{\tau}$ is unbiased, and its mean squared error reduces to
> $$\mathbb{E}\big[(\hat{\tau}-\tau)^2\big] = \operatorname{Var}(\hat{\tau}) = \frac{\sigma^2(1)}{n(1)} + \frac{\sigma^2(0)}{n(0)}.$$
>
> Therefore, ATE estimation is essentially a two-group mean estimation problem for $p=1$ norm, where the key design question is how to allocate samples across groups to minimize estimation error. This is closely aligned with our objective. We refer the reviewer to [3,4] for the background and problem setting on adaptive Neyman allocation and experimental design, and to [5,6] for representative applications in the social science literature.
>
> We thank the reviewer again for the questions and suggestions. We are happy to have further discussions in the following week.
>
> [1] Aznag, A. et al. An active learning framework for multi-group mean estimation.
>
> [2] Carpentier, A. et al. Upper-confidence-bound algorithms for active learning in multi-armed bandits.
>
> [3] Zhao, J. Adaptive neyman allocation.
>
> [4] Li, J. et al. Optimal adaptive experimental design for estimating treatment effect.
>
> [5] Blackwell, M. et al. Batch adaptive designs to improve efficiency in social science experiments.
>
> [6] Xu, J. et al. MUSE: Multi-Treatment Experiment Design for Winner Selection and Effect Estimation.

---

> > ### Author Rebuttal · Reviewer_8h7i · 2026-04-03
> >
> > the authors have done a good job addressing my concerns, i have updated my scores

---

### Decision · Program_Chairs · 2026-04-30

**Decision:**

Accept (regular)

**Comment:**

The paper studies a bandit problem in which the goal is to estimate the means of all arms with uniform accuracy, when the arms may have unknown and heterogeneous variances. The paper derives the optimal number of arm pulls and proposes two algorithms: one that assumes knowledge of a lower bound on the variances and an upper bound on the sub-Gaussian parameter, and another that uses only the latter. Both algorithms are “exploration-free” in the sense that the number of pulls required to obtain a sufficiently accurate variance estimate is smaller than the number of arm pulls in the optimal allocation scheme. Both also achieve tighter regret bounds than prior work. The proposed method is further extended to the contextual linear bandit setting, where the goal is to estimate the arm-specific linear parameter.

Reviewers overall agree that the problem is well-motivated and that the theoretical development of the algorithm is sound. There were some concerns regarding the practical relevance of the proposed framework, which the authors effectively addressed through the average treatment effect estimation example. The authors are strongly encouraged to include this example in the final version. The authors are also required to include an Impact Statement in the final version.